# Enhancement of Ambulatory Glucose Profile for Decision Assistance and Treatment Adjustments

**DOI:** 10.3390/diagnostics14040436

**Published:** 2024-02-16

**Authors:** V. K. R. Rajeswari Satuluri, Vijayakumar Ponnusamy

**Affiliations:** Department of ECE, SRM Institute of Science and Technology, Kattankulathur 603203, Tamil Nadu, India; rs3740@srmist.edu.in

**Keywords:** CGM device, clinical metrics, decision assistive system, standardized metrics

## Abstract

The ambulatory glucose profile (AGP) lacks sufficient statistical metrics and insightful graphs; indeed, it is missing important information on the temporal patterns of glucose variations. The AGP graph is difficult to interpret due to the overlapping metrics and fluctuations in glucose levels over 14 days. The objective of this proposed work is to overcome these challenges, specifically the lack of insightful information and difficulty in interpreting AGP graphs, to create a platform for decision assistance. The present work proposes 20 findings built from decision rules that were developed from a combination of AGP metrics and additional statistical metrics, which have the potential to identify patterns and insightful information on hyperglycemia and hypoglycemia. The “CGM Trace” webpage was developed, in which insightful metrics and graphical representations can be used to make inferences regarding the glucose data of any user. However, doctors (endocrinologists) can access the “Findings” tab for a summarized presentation of their patients’ glycemic control. The findings were implemented for 67 patients’ data, in which the data of 15 patients were collected from a clinical study and the data of 52 patients were gathered from a public dataset. The findings were validated by means of MANOVA (multivariate analysis of variance), wherein a *p* value of < 0.05 was obtained, depicting a strong significant correlation between the findings and the metrics. The proposed work from “CGM Trace” offers a deeper understanding of the CGM data, enhancing AGP reports for doctors to make treatment adjustments based on insightful information and hidden patterns for better diabetic management.

## 1. Introduction

The AGP is a comprehensive report of a patient’s summarized glycemic control, in the form of statistical and graphical representations generated from continuous glucose monitoring (CGM) data. The AGP report comprises statistical data, including the average/mean glucose (MG), the glycemic variability (GV), the time in range (TIR), the time above range (TAR), the time below range (TBR), and the graphical metrics depicting the median and target levels of BG, along with a graph of the patient’s daily glucose profile over 14 days [1,2,3].

The statistics, graphs, and predefined glucose target ranges of the AGP report help doctors to make informed decisions and provide personalized guidance to patients for better diabetic management. Researchers employing the AGP report have stated its limitations for interpretation and the need for other valuable metrics [4,5,6,7]. Insightful metrics in decision-support tools may allow clinicians to make better decisions, strategize, adjust patients’ treatments to more precise drug dosages, and better track their patients’ health. Additionally, patient welfare may be enhanced, and patients may be more likely to adhere to a treatment regimen. In the related research, it was found that the GV is strongly correlated with intraday and interday glucose variations, where day-to-day data and data for ≤14 days are important for identifying fluctuations; however, this is not generated in AGP reports [8]. The glucose management indicator (GMI) is a measure of approximate HbA1C values, based on an average glucose level over 14 days. It is also mentioned that the GMI and the laboratory-tested HbA1C values are different, as the AGP report does not consider non-glycemic factors, such as the survival time of erythrocytes and hemoglobin, etc. Hyperglycemia/TAR, the time below hypoglycemia/(TBR), and the time in range (TIR) can be used to assess the risk of dysglycemia/prediabetes. Patient trends, patterns of GV, response to medication, and persistence, as well as the frequency, severity, duration, and recurrences of hypoglycemia and hyperglycemia, are crucial in identifying clinical problems and providing solutions for patients [8,9,10]. In the summarized interpretation of the AGP report, a patient’s GV at their lowest glucose level is diluted below 10% [9], leaving out important information on acute glucose fluctuations [8,9,11], which can be appropriately evaluated by the mean amplitude of glycemic excursions (MAGE) [12]. The MAGE is a numerical measure of an average amplitude of upward and downward glucose excursions. In the AGP report, analyzing day-to-day inconsistencies in meal timings, exercise, and insulin administration is not possible if there are less than 14 days recorded, which fails to take into account cases of dysglycemia/prediabetes [9,10,13]. The MG, TIR, and GV values in the AGP report are insufficient to infer any insight from interstitial glucose levels. The research gaps identified from the existing AGP report are as follows:The existing metrics are insufficient to analyze a complete glucose profile, which fails to capture the temporal patterns of glucose fluctuations that may lead to increased risks of hyperglycemia and hypoglycemia, as well as cardiovascular, neuropathic, and nephropathic issues.With just the combination of the MG, GV, TIR, and GMI values and the graph from the AGP report, deeper insights into patients’ glucose profiles are not impossible, such as their interday GV, intraday GV, and glucose fluctuations, resulting in incomplete assessments and poor diabetes control.The AGP report is a 14-day comprehensive view of the effect of a treatment regimen and lifestyle changes, including patients’ meal intake and activities; however, it is crucial to analyze glucose data every day based on the severity of complications and during emergencies.Patients’ response to medication, the persistence of glycemic peaks or lows, and reoccurrences of hyperglycemia and hypoglycemia are crucial factors that are hard to analyze using the metrics and the complex graphical representation in the AGP report.

There is a need for a report on combined statistical metrics to find insights with a combination of metrics and graphical trends to accurately detect hyperglycemia, hypoglycemia, hidden patterns of hyperglycemia and hypoglycemia, or hyperglycemia and hypoglycemia together at specific intervals. Moreover, contextual information, i.e., patients’ response to medication changes, the effect of illnesses, meal plans, and physical activity, cannot be analyzed using a summarized 14-day report, which is a disadvantage. To address these issues, the development and management of decision support tools to assist users are briefly outlined for the maximal use of CGM, and an attempt is made to enhance the AGP report [11,12]. This is our motivation to develop “CGM Trace” for better diabetic management. The contributions of the proposed work are as follows:Decision rules are formulated with a combined assessment of metrics to assist doctors in gaining insights into glucose variability, temporal patterns of glucose fluctuations, patients’ response to medication, persistence in glycemic peaks or lows, and reoccurrences of hyperglycemia and hypoglycemia to create accurate treatment regimens.This work presents the findings from the five standard AGP metrics, the five added statistical metrics, and the five graphical trends in a place that lacks an AGP report on the “CGM Trace” dashboard system.The decision rules are statistically validated using MANOVA, in which all metrics obtained a *p* value of <0.05, implying a significant correlation between the metrics and the findings.“CGM Trace” enhances the AGP report by generating five graphical presentations for identifying glucose trends and patterns, providing a distinct view of glycemic excursions.

The current work is organized in the following manner: Section 2 presents the Materials and Methods, where CGM data collection, the metrics incorporated into “CGM Trace”, and the proposed decision rules are discussed. Section 3 presents the validation of the decision rules, which was performed using MANOVA, analyzing the results of the web interface “CGM Trace”, and MANOVA implementation in 67 patients. Section 4 presents a discussion on the findings found from the clinical dataset and the public dataset. This study ends with an exposition on the conclusion.

## 2. Materials and Methods

This section describes the data collection, added metrics, and proposed decision rules. The added statistical metrics and graphical representations are incorporated into “CGM Trace” for gaining maximum benefits from the CGM data when compared with the AGP report.

### 2.1. CGM Data Collection

Ethical clearance was obtained from the SRM Medical College Hospital and Research Centre, Kattankulathur-603203, Tamil Nadu, India (Ethical clearance number: 8274/IEC/2022). A publicly available “closed-loop control to range system” public dataset was also obtained from the JCHR-JAEB Center for Health Research, which was the coordinating center. The Inpatient Evaluation of an Automated Closed-Loop Control-to-Range System (*CTR*) (NCT01271023) (Inpatient Evaluation of an Automated Closed-Loop Control-to-Range System) was retrieved from https://public.jaeb.org/datasets/diabetes. The analysis, content, and conclusions presented herein are solely the responsibility of the authors and have not been reviewed or approved by the Jaeb Center for Health Research/Inpatient Evaluation of an Automated Closed-Loop Control-to-Range System (CTR). The full protocol is available online “www.clinicaltrials.gov/ct2/show/NCT01271023 (accessed on 12 June 2023)”. Written informed consent was obtained from each patient or parent, with assent obtained as required. This study was designed and conducted according to the ethical principles that comply with the Declaration of Helsinki. In this work, clinical study data originated from Tamil Nadu, India, USA, France, Italy, and Israel, collected by JCHR JAEB. Patient data anonymization was strictly performed by omitting the patient’s name, address, and other personal details. The dataset for the proposed work was created considering the date, time, and glucose levels.

In this work, 67 patients, including 15 patients from the clinical study and 52 patients from the public dataset, were considered. Patient demographics are detailed in Table 1. Patients with diabetes on CGM sensors with no mental health and cognitive disorders were chosen for this study. Pregnant and lactating women and patients with diabetic ketoacidosis, seizure disorders, active infection, muscular conditions, cancer, or cystic fibrosis were excluded due to the possibility of bias. 

### 2.2. Metrics Incorporated into “CGM Trace”

“CGM Trace” was developed by incorporating five standard AGP metrics and five added statistical metrics along with five graphical trends, as illustrated in Figure 1. As the user (doctor and caregiver/patient) uploads a .csv file, the user can view the numeric and graphical representations, from which they can infer the interstitial glucose status. However, the doctor has access to decision rules in the form of “Major Findings”, along with numeric and graphical representations, which are illustrated as time in ranges, glucose statistics, and the glucose profile. The 20 proposed decision rules implied the findings associated with the rules. The findings were: hyperglycemia with the rate of change in glucose levels; high glycemic variability and high glycemic fluctuations; hyperglycemia with a high rate of change in glucose levels; hyperglycemia with high glycemic variability; hyperglycemia with high glycemic fluctuations; hypoglycemia with a high rate of change in glucose levels; high glycemic variability and high glycemic fluctuations; hypoglycemia with a high rate of change in glucose levels; hypoglycemia with high glycemic variability; hypoglycemia with high glycemic fluctuations; hidden hyperglycemia; hidden hyperglycemia with a high rate of change in glucose levels; hidden hyperglycemia with high glycemic variability; hidden hyperglycemia with high glycemic fluctuations; hidden hypoglycemia; hidden hypoglycemia with a high rate of change in glucose levels; hidden hypoglycemia with high glycemic variability; hidden hypoglycemia with high glycemic fluctuations; hyperglycemia and hypoglycemia; hyperglycemia and hypoglycemia with a high rate of change in glucose levels; hyperglycemia and hypoglycemia with high glycemic variability; and hyperglycemia and hypoglycemia with high glycemic fluctuations. The proposed “CGM Trace” is a universal software tool developed by implementing standardized metrics. Irrespective of the origin of the data, meals and physical activity are reflected in the glucose levels, which are thoroughly analyzed using “CGM Trace”.

#### Added Metrics

The proposed statistical metrics are beneficial for understanding the CGM data. Incorporating these factors is beneficial for determining glucose trends and behaviors. The following are the proposed metrics. Statistical formulas for evaluating the metrics are provided in Appendix A.
The standard deviation rate of change (SDR): The SDR is defined as the standard deviation (SD) calculated on the rate of change of glucose levels [14,15,16]. As the SD is highly asymmetric, with an unequal distribution of hypoglycemic and hyperglycemic ranges leading toward biased assessment, the SDR is advantageous in indicating fluctuations and higher variability in glucose levels over time. The CGM sensor collects data every 5 min; therefore, the SDR is computed with SD over 5 min. An SDR of 5 is a threshold to indicate glucose levels with higher fluctuations [17]. The metric SDR is helpful for the doctor to identify hypoglycemia [14] and it is a crucial tool for predicting post-prandial GV. It is an important tool to make informed decisions regarding insulin dosage [15]. Correlating SDR with TIR and TAR helps in identifying chronic kidney disease, myocardial infarction, heart failure, and stroke [18]. Incorporating SDR into “CGM Trace” will provide information about the glucose levels’ improvement, deterioration, or stability, for effective decision-making and treatment adjustments of increasing, decreasing, or continuing the exact drug dosage.The interquartile range (IQR): The non-uniform distribution of a hypoglycemic and hyperglycemic range of the glucose margin creates an asymmetrical distribution, where the IQR is more important than the SD for non-symmetrical distributions [19]. Incorporating the IQR into “CGM Trace” will provide insight for the doctor to understand the spread of glucose levels visually. A wider IQR indicates a spread of glucose, indicating instability and inconsistency, thus implying a need to improve/increase the drug dosage for glycemic control.The hypoglycemic events graph: Hypoglycemia is a condition of low blood glucose levels when blood glucose is <70 mg/dL [20]. Incorporating a hypoglycemic graph into “CGM Trace” is beneficial for the doctor to identify the events of hypoglycemia and its specific hidden patterns, which cannot be inferred from the AGP report. It will provide the doctor with a vision of the nocturnal lows, information on the effect of insulin, medications, meal plans, physical activity, and underlying diseases, such as high arterial stiffness risk of albuminuria, retinopathy, cardiovascular disease mortality, all-cause mortality, and abnormal carotid intima-media thickness [18,20,21,22].The hyperglycemic events graph: Hyperglycemia is a condition of high blood glucose levels when blood glucose is >180 mg/dL [19,23]. Incorporating hyperglycemic events into “CGM Trace” will help the doctor to assess the post-meal spikes, glucose concentration, effectiveness of drug dosage, uncontrolled diabetes, decisions for new treatment approaches, and identification of microvascular and macrovascular complications [24,25].The intraday glucose trend: Intraday is the glucose trend divided into bolus infusion timings, that is, during the morning, afternoon, and at midnight. The intraday trend represents the glucose concentration through the Poincaré plot [19,23]. Incorporating the intraday glucose trend into “CGM Trace” will allow the doctor to make an informed decision on the drug dosage and correct the dose of insulin depending on factors such as recurring patterns, meal intake, response to drugs, and physical activity. It can assist the doctor in recommending meal plans based on post-meal spikes or midnight lows.The overall glucose trend: The overall glucose trend represents interstitial glucose in a selected time series [19]. Incorporating the intraday glucose trend into “CGM Trace” will allow doctors to decide on treatment adjustments by analyzing the glycemic control over some time. Assistance regarding meal plans and treatment adjustments can be provided based on the number of times the patient has achieved target glucose levels (<180 mg/dL), post-meal spikes, and midnight lows.The standard deviation (SD): A measure of the spread of glucose obtained around the average, where an SD < 33 is desirable [20]. Incorporating SD into “CGM Trace” will allow the doctor to identify the higher or lower glucose variability and decide on changes in the treatment regimen.The mean of daily differences (MODD): A measure of the interday GV estimation. It is calculated as the mean of the absolute differences between glucose concentrations measured at the same time of the day for two consecutive days [19]. Incorporating MODD into “CGM Trace” will allow clinicians to decide on drug dosage based on the food consumed and the bolus insulin/medications received. There is no threshold for MODD; however, a higher MODD is indicative of irregular food and lifestyle habits [24]. The MODD of the patients can be compared from the last visit to the current visit to identify the improvement in the patient’s lifestyle and diabetic management. It helps the doctor to assist patients with proper eating and lifestyle habits to achieve a lower MODD and to identify microvascular and macrovascular complications [25,26].The continuous overall net glycemic action (CONGA): The CONGA is the SD measured as the difference between a current glycemic observation and another observation n hours apart [19]. Incorporating CONGA into “CGM Trace” will allow doctors to make data-driven decisions based on the food consumed and the bolus insulin/medications received. There is no threshold for CONGA; however, it was observed that CONGA increased gradually from 1 to 8 h in the case of higher variability in glucose. It was also observed to be stable at 4 h if the glucose level is not fluctuating [27]. The CONGA of the patients can be compared from the last visit to the current visit to identify the improvement in the patient’s lifestyle and diabetic management. It will also enable doctors to predict long-term hyperglycemic or hypoglycemic events, make treatment adjustments, and reduce suboptimal glucose control, which is related to identifying microvascular and macrovascular complications [24,26].The mean amplitude of glycemic excursions (MAGE): Calculated as the arithmetic mean of the differences between the consecutive peaks and troughs of differences greater than one time the SD of the mean glycemia. It estimates major glucose swings and excludes minor swings [2,25]. There is no threshold for MAGE; however, 40 mg/dL was found in a clinical study to be indicative of suboptimal glucose control and was considered as a reference in this work [28]. Incorporating MAGE into “CGM Trace” will allow clinicians to analyze the glycemic excursions to identify specific periods of peaks and lows for creating a tailored treatment plan to achieve target glucose levels.

### 2.3. Proposed Decision Rules

The proposed decision rules were developed from the combined assessment of all metrics. Glycemic control was considered normal when the metrics were within the target levels, i.e., MG = 70–180 mg/dL, TIR > 70%, TAR-I < 25%, TAR-II < 5%, TBR-I < 4%, and TBR-II < 1% [2,29,30]. However, metrics such as SDR < 5, SD < 33, GV ≤ 36, and MAGE < 40 from various studies implied the presence of glycemic variability, instability, and fluctuations at a given time [2,17,28,29,30]. The combined metrics with MAGE and CONGA can be analyzed for identifying glycemic control over specific time intervals and MODD for its variability. A centered IQR, with intraday and overall trends, provides information on the spread of glucose, its variability, and its peaks and lows [28,31,32,33,34]. Decision rules were formed from the combined literature and clinical studies categorized based on poor and unstable glycemic control, as presented in Table 2. The decision rules were categorized into hyperglycemia, hypoglycemia, hidden hyperglycemia, hidden hypoglycemia, hyperglycemia, and hypoglycemia. Hidden hyperglycemia is a condition when either or both of TAR-I > 25% and TAR-II > 5% are above the target levels, even when the overall glucose levels are within the target range (MG < 180, TIR > 70%, GV ≤ 36, SDR < 5, MAGE < 40). Similarly, hidden hypoglycemia is a condition when either or both of TBR-I > 4% and TBR-II > 1% are above the target levels, even when the overall glucose levels are within the target range (MG < 180, TIR > 70%, GV ≤ 36, SDR < 5, MAGE < 40).

R1 applies to the patient when MG, TAR-I, TAR-II, GV, SDR, and MAGE are above the target levels and TIR is below the target level, indicating hyperglycemia. R2 applies to the patient when MG, TAR-I, TAR-II, and SDR are above the target levels, TIR is below the target level, and GV and MAGE are at the target levels, indicating hyperglycemia with a high rate of change in glucose. R3 applies to the patient when MG, TAR-I, TAR-II, and GV are above the target levels, TIR is below the target level, and SDR and MAGE are at the target levels, indicating hyperglycemia with high glycemic variability. R4 is applicable to the patient when MG, TAR-I, TAR-II, and MAGE are above the target levels, TIR is below the target level, and SDR and GV are at the target levels, indicating hyperglycemia with high glucose fluctuations.

R5 applies to the patient when TBR-I, TBR-II, GV, SDR, and MAGE are above the target levels and MG and TIR are below the target levels, indicating hypoglycemia. R6 applies to the patient when TBR-I, TBR-II, and SDR are above the target levels, MG, and TIR are below the target levels, and GV and MAGE are within the target levels, indicating hypoglycemia with a high rate of change in glucose. R7 applies to the patient when TBR-I, TBR-II, and GV are above the target levels, MG and TIR are below the target levels, and SDR and MAGE are within the target levels, indicating hypoglycemia with high glycemic variability. R8 applies to the patient when TBR-I, TBR-II, and MAGE are above the target levels, MG and TIR are below the target levels, and SDR and GV are at the target levels, indicating hypoglycemia with high glucose fluctuations.

R9, as hidden hyperglycemia, is applicable when the overall glucose metrics, i.e., MG, GV, SDR, MAGE, and TIR, are within the target levels but both or either (mentioned as OR in the decision rules from Table 2) of TAR-I and TAR-II in the decision rules are above the target glucose levels, indicating events of hyperglycemia under normal overall glucose levels. R10, as hidden hyperglycemia with a high rate of change in glucose levels, is applicable when the overall glucose metrics, i.e., MG, GV, MAGE, and TIR, are within the target levels but both or either of TAR-I and TAR-II with SDR are above the target glucose levels. R11, as hidden hyperglycemia with high glycemic variability in glucose levels, is applicable when the overall glucose metrics, i.e., MG, MAGE, SDR, and TIR, are within target levels and both or either of TAR-I and TAR-II with GV are above the target glucose levels. R12, as hidden hyperglycemia with high glucose fluctuations in glucose levels, is applicable when overall glucose metrics, i.e., MG, GV, SDR, and TIR, are within the target levels but both or either of TAR-I and TAR-II with MAGE are above the target glucose levels.

R13, as hidden hypoglycemia, is applicable when the overall glucose metrics, i.e., MG, GV, SDR, MAGE, and TIR, are within the target levels but both or either of TBR-I and TBR-II are above the target glucose levels, indicating events of hypoglycemia under normal overall glucose levels. R14, as hidden hypoglycemia with a high rate of change in glucose levels, is applicable when the overall glucose metrics, i.e., MG, GV, MAGE, and TIR, are within the target levels but both or either of TBR-I and TBR-II with SDR are above the target glucose levels. R15, as hidden hypoglycemia with high glycemic variability in glucose levels, is applicable when the overall glucose metrics, i.e., MG, MAGE, SDR, and TIR, are within the target levels but both or either of TBR-I and TBR-II with GV are above the target glucose levels. R16, as hidden hypoglycemia with high glucose fluctuations in glucose levels, is applicable when the overall glucose metrics, i.e., MG, GV, SDR, and TIR, are within the target levels but both or either of TBR-I and TBR-II with MAGE are above the target glucose levels.

The patient’s condition is assessed with R17 when both or either of TAR-I and TAR-II and both or either of TBR-I and TBR-II are above the target ranges, indicating hyperglycemia and hypoglycemia occurring in different intervals in the events of hyperglycemic spikes and hypoglycemic episodes [35,36,37]. R18 is assessed in the patient when both or either of TAR-I and TAR-II and both or either of TBR-I and TBR-II with SDR are above the target ranges, indicating the occurrence of hyperglycemia and hypoglycemia at different intervals with a high rate of change in glucose levels. R19 is assessed in the patient when both or either of TAR-I and TAR-II and both or either of TBR-I and TBR-II with GV are above the target ranges, indicating the occurrence of hyperglycemia and hypoglycemia at different intervals with high glycemic variability. R20 is assessed in the patient when both or either of TAR-I and TAR-II and both or either of TBR-I and TBR-II with MAGE are above the target ranges, indicating the occurrence of hyperglycemia and hypoglycemia at different intervals with high glucose fluctuations. Each finding differs with a combination of TAR-I, TAR-II, TIR, TBR-I, and TBR-II with SDR, GV, and MAGE, which are above, below, or within the target levels.

However, there is a possibility of a combination of decision rules. When the patient’s condition is a combination of any two conditions between the high rate of change in glucose, high glycemic variability, and high glucose fluctuations, the decision rules are aggregated and displayed; for example, in the case of hyperglycemia with a high rate of change in glucose, high glycemic variability is programmed to aggregate (R2 + R3) and display.

## 3. Results

This section presents the validation of the decision rules, the web interface, and the data analysis from “CGM Trace”. MANOVA one-way test is also presented with the assumption given in Table 3.The decision rules were evaluated in 67 patients from “CGM Trace”, of which the data for 15 patients from the clinical study are presented in Table 4 and the data for the first 15 patients from the public dataset are presented in Table 5. Due to the space concern in this study, the evaluated metrics on the remaining 37 patients from public data are provided in Appendix A.

### 3.1. Validation of Decision Rules

In this work, the data consisted of one factor, i.e., the findings, and multiple dependent metrics. Statistical analysis using one-way MANOVA was performed using IBM SPSS Statistics 27 software. The statistical significance of the *p*-value was set at *p* < 0.05 [38,39]. The normality of the data is the fundamental assumption of MANOVA. The assumptions and *p*-value from the MANOVA one-way test are described in Table 3. As the data were skewed, an inverse distribution function (DF) was performed to transform the data into a normal distribution. The second assumption of MANOVA is that the covariances of matrices must be equal, which was achieved with *p* = 0.531 (rejecting the null hypothesis that the covariances of the matrices are not equal). The findings were the factor with 20 levels, i.e., F1-F20, which were the independent variables. The dependent variables were the metrics, i.e., SDR, MG, GV (%), TIR (%), TAR-I (%), TAR-II (%), TBR-I (%), TBR-II (%), and MAGE. From the tests of the between-subject effects, *p* = 0.791 was achieved, which stated that the independent variables affected the dependent variables (rejecting the null hypothesis that there was no effect of the independent variables on the dependent variables), satisfying the third assumption of MANOVA. CONGA and MODD were omitted from the decision rules as they do not have thresholds; however, SDR was chosen over SD due to its asymmetric nature, with unequal distribution of hypoglycemic and hyperglycemic ranges leading toward biased assessment. SD, CONGA, and MODD were considered as supporting metrics for data analysis. A null hypothesis was formed, stating that the findings would not differ and had no significance among all the dependent metrics. The overall significance obtained between the findings and the metrics was *p* = 0.001 using Wilk’s lambda test, where a partial eta-squared value of 0.395 was acquired. It can be interpreted that there was a difference and a strong significance between the findings and the metrics at *p* < 0.05, rejecting the null hypothesis.

### 3.2. Web Interface and Data Analyzation from “CGM Trace”

Different combinations of decision rules in identifying hyperglycemia and hypoglycemia were formed based on the literature, as illustrated in Table 2. Decision rules were programmed into the software of “CGM Trace”. The web page was designed using Python Flask, CSS, and JavaScript. The user could upload the CSV file and select a timeframe to view the numerical metrics, as illustrated in Figure 2, and trends metrics, as depicted in Appendix A. The doctor could access the findings to view a comprehensive report on glycemic control along with metrics and graphs, as shown in Figure 3. The findings were displayed at the top of the web page, highlighting the summary of the patient’s condition. Time in ranges provided a clear insight into TIR, TAR, and TBR with glucose statistics. A distinct view of glycemic control could be accessed through a graphical representation from “Glucose Profile” to analyze the effect of treatment, changes in medication, physical activity, and meals, which could be easily accessed in case of emergency.

The CGM dataset consisted of date, time, and glucose measurements. The file was uploaded into “CGM Trace” and the metrics were evaluated for 67 patients, which are illustrated in Table 4 for the clinical data, Table 5 for the public data, and S1 for the continuation of 37 patients for data analysis.

The validated decision rules, i.e., (R1-20) from Table 2, were assigned to each patient in Table 4 and Table 5, and S1 for data analysis. Cases of hyperglycemia, hypoglycemia, hidden hyperglycemia, hidden hypoglycemia, hyperglycemia, and hypoglycemia are discussed thoroughly.

Evaluating glucose levels for hyperglycemia with only MG and TIR from the AGP report cannot reflect hyperglycemia and its associations with other complications, leading to erroneous assessments. Elevated glucose levels, where SDR > 5, MG > 180, GV > 36, TIR < 70%, TAR-I > 25%, TAR-II > 5%, and MAGE > 40 mg/dL, are associated with hyperglycemia [15,26,28,30,32,33,34,40,41,42]. This can be observed in patients 11 and 13 from Table 4, patient 15 from Table 5, and patients 1, 3, and 31 from S1, where R1 indicates hyperglycemia with a high rate of change in glucose, high glycemic variability, and high glucose fluctuations are applicable. Normal GV and MAGE with TIR < 70%, TAR-I > 25%, TAR-II > 5%, and SDR > 5 can be observed in patients 1 and 5 from Table 5 and patients 17, 20, 28, and 29 from S1, indicating hyperglycemia with a high rate of change in glucose where R2 is applicable. Similarly, normal SDR and MAGE with MG > 180 mg/dL, TIR < 70%, TAR-I > 25%, TAR-II > 5%, and GV > 36 can be observed in patient 14 from Table 5 and patient 37 from S1 where R3 is applicable, indicating hyperglycemia with a high glucose variability. TIR < 70%, TAR-I > 25%, TAR-II >5%, normal SDR and GV, and MAGE < 40 and can be observed in patients 11, 24, 27, and 30 from S1, indicating R4 hyperglycemia with high glucose fluctuations. In patient 8 from Table 5, TIR < 70%, TAR-I > 25%, and TAR-II > 5% can be observed with SDR > 5 and MAGE > 40. In this case, the decision rules are combined from R2 and R4, indicating hyperglycemia with a high rate of change in glucose, and high glucose fluctuations. Similarly, in patient 22 from S1, TIR < 70%, TAR-I > 25%, and TAR-II > 5% can be observed with GV > 36 and MAGE > 40, where R3 and R4 were combined into a single finding, indicating hyperglycemia with high glycemic variability and high glucose fluctuations. As SD, IQR, MODD, and CONGA are correlated, the hyperglycemic state can be analyzed using the metrics. With an SD > 33, the variability in glucose levels can be observed from MODD. Variability in glucose levels can be observed from CONGA with the combined metrics. The IQR graph can be analyzed where a wide IQR range depicts the spread of glucose in patients with hyperglycemia, as shown in Figure 4 for patient 30. It can be inferred from the IQR graph that the spread of glucose ranged from 190 mg/dL to 263 mg/dL. Similarly, intraday glucose trends in Figure 5 of patient 30 provide precise insights into glucose concentrations in the morning, afternoon, and night, where glucose > 180 mg/dL. The IQR intraday glucose trends and hyperglycemic trends graphs provide deeper insights into glucose oscillations for selected periods (from hours to days) when compared with the AGP graph.

MG and TIR alone from the AGP report cannot emphasize hypoglycemia and its correlating diseases, leading to inaccurate assessments. Hypoglycemia can be precisely identified using combined metrics where MG < 180 mg/dL, TBR-I > 4%, TBR-II > 1%, TIR < 70%, GV > 36, SDR > 5, and MAGE > 40 [15,19,28,30,32,33,35,43,44,45,46]. This can be observed in patient 5 from Table 4, where R5 indicates hypoglycemia with a high rate of change in glucose, high glycemic variability, and high glucose fluctuations. R6 is applicable where normal GV and MAGE with TBR-I > 4%, TBR-II > 1%, TIR < 70, and SDR > 5, and can be observed in patient 6 from Table 4 and patient 12 from S1, indicating hypoglycemia with a high rate of change in glucose. TBR-I > 4%, TBR-II > 1%, TIR < 70%, and GV > 36 with normal SDR and MAGE and can be observed in patient 7 from Table 4, indicating hypoglycemia with high glucose variability where R7 is applicable. TBR-I > 4%, TBR-II > 1%, TIR < 70%, and MAGE > 40 with normal SDR and GV can be observed in patient 10 from Table 5, and patient 32 from S1, where R8 indicating hypoglycemia with high glucose fluctuations is applicable. Normal GV with TBR-I > 4%, TBR-II > 1%, TIR < 70%, SDR > 5, and MAGE > 40 can be observed in patients 1 and 2 from Table 4, indicating hypoglycemia with a high rate of change in glucose, and high glucose fluctuations where R6 and R8 are applied in combination. In patient 10 from Table 5, SD > 33 and (morning, afternoon, and midnight) CONGA can be compared with the combined metrics to infer the severity of glycemic variability. A hypoglycemic state can be inferred from the hypoglycemic trend graph, as depicted in Figure 6A, and intraday glycemic trend graphs for reviewing its trends at specific time intervals (morning, afternoon, and midnight) can be collected from “CGM Trace”. With SD > 33, the variability in glucose levels can be observed from MODD. Variability in glucose levels can be observed from CONGA using the combined metrics.

Hidden hyperglycemia implies elevated glucose levels at a certain time of the day, even when the overall glucose levels fall within the target range. These can be identified as peaks for a few hours and post-meal spikes, even under normal MG, TIR, TAR-I, TAR-II, SDR, GV, and MAGE. Hidden hyperglycemia cannot be identified from the AGP report, which leaves out crucial information on glucose levels. Normal MG, TIR, SDR, GV, and MAGE with TAR-II > 5% can be identified in patient 8 from Table 4, indicating hidden hyperglycemia where R9 is applicable. Normal SDR, TAR-I, TIR, and MAGE with GV > 36 can be observed in patient 19 from S1, where R11 indicating hidden hyperglycemia with high glycemic variability is applicable. Similarly, normal TAR-I, TIR, SDR, and GV with TAR-II > 5% and MAGE > 40 can be observed in patient 33 from S1, where R12 indicating hidden hyperglycemia with high glucose fluctuations is applicable. Normal TAR-I, TIR, and SDR with TAR-II > 5%, GV > 36, and MAGE > 40 can be observed in patient 5 from S1, indicating hidden hyperglycemia with high glucose variability, and high glucose fluctuations where R11 and R12 are applied in combination. MODD and CONGA in patients can be compared with the combined metrics. In patient 5, who had an MODD of 7.4 and a CONGA of 8.2 × 10^−16^ with SD > 33, MAGE > 40 was observed, depicting higher glycemic variability and high glucose fluctuations with hyperglycemia. The patterns of hidden hyperglycemia can be found in intraday glucose trends from “CGM Trace”, which focus on glucose elevations at three times of the day. The hyperglycemic trend graph can also be analyzed to identify high glucose levels precisely. Though TIR > 70% can be identified in these patients, the evidence for the patients with above target glucose levels indicates hidden hyperglycemia.

Hidden hypoglycemia is low glucose levels at a certain time of the day, even when the overall glucose levels fall within the target range. These can be identified as nocturnal or sudden lows in glucose levels that are not apparent immediately, which is impossible to identify in the AGP report, leading to erroneous assessments of interstitial glucose. Normal MG, SDR, TIR, GV, and MAGE can be found in patients 8 and 34 from S1.

TBR-I > 4% or TBR-II > 1% was identified in these patients, where R13 indicating hidden hypoglycemia is applicable. Normal MG, TIR, GV, and MAGE with TBR-I > 4% and SDR > 5 can be identified in patient 35 from S1, indicating hidden hypoglycemia with a high rate of change in glucose level where R14 is applicable. In patients 14 and 18, normal SDR, TIR, MG, and MAGE were identified with GV > 36 from S1. TBR-I > 4% or TBR-II > 1% were identified in these patients, indicating hidden hypoglycemia with high glycemic variability where R15 is applicable. Normal MG, TIR, GV, and SDR with MAGE > 40 can be identified in patient 3 from Table 4 and in patients 4 and 14 from Table 5, indicating hidden hypoglycemia with high glucose fluctuations where R16 is applicable. Normal TIR and MAGE with TBR-I > 4%, TBR-II > 1%, SDR > 5, and GV > 36 can be observed in patient 9 from S1 where R14 and R15 are applied in combination, indicating hidden hypoglycemia with a high rate of change in glucose and high glycemic variability. In patient 3, who had an MODD of 6.7 and a CONGA of 8.8 × 10^−16^ with SD > 33, MAGE > 40 was observed, depicting higher glycemic variability and high glucose fluctuations with hidden hypoglycemia. The patterns of hidden hypoglycemia can be found in intraday glucose trends from “CGM Trace”, which provides insights on glucose spikes at specific times of the day. A hypoglycemic trend graph can be inferred for identifying the precise trend analysis. Though TIR > 70% can be identified in these patients, the evidence for the patients with below target glucose levels indicates hidden hypoglycemia. Treatment changes to avoid hidden hypoglycemia are required for addressing hypoglycemic fluctuations in glucose levels, to avoid adversities due to coma or death. Coronary artery disease is observed to be correlated with hypoglycemic events and lows in intraday glycemic variability [26,47].

Hyperglycemia and hypoglycemia coexist and occur in a day as post-prandial highs or lows, along with nocturnal lows of glucose levels at different intervals. It is difficult to identify hyperglycemia and hypoglycemia with a single metric of MG and TIR in the AGP report. The graphical trends for 14 days in the AGP report fail to provide insights into the finer points of the lows and highs of glucose. Normal GV and MAGE with TIR < 70%, TAR-I > 25% or TAR-II > 5%, TBR-I > 4% or TBR > 1%, and SDR > 5 can be identified in patient 16 from S1 where R18 is applicable, indicating hyperglycemia and hypoglycemia with a high rate of change in glucose. Normal SDR, MAGE with TIR < 70%, TAR-I > 25% or TAR-II > 5%, TBR-I > 4% or TBR > 1%, and GV > 36 can be identified in patients 2 and 11 from Table 5 and patient 7 from S1, indicating hyperglycemia and hypoglycemia with high glycemic variability where R19 is applicable. For TIR < 70%, TAR-I > 25% or TAR-II > 5%, TBR-I > 4% or TBR > 1%, and MAGE > 40 with normal SDR, GV can be identified in patient 3 from Table 5 and patient 15 from S1, where R20 indicating the case of hyperglycemia and hypoglycemia with high glucose fluctuations is applicable. TIR < 70%, TAR-I > 25%, TAR-II > 5%, TBR > 1%, GV > 36, MAGE > 40, and normal SDR were identified in patient 12 from Table 5, where R19 and R20 are applied, indicating hyperglycemia and hypoglycemia with high glycemic variability and high glucose fluctuations. TIR < 70%, TAR-II > 5%, TBR I > 4%, TBR > 1%, GV > 36, MAGE > 40, and SDR > 5 were identified in patients 2 and 23 from S1.

In this case, the decision rules were combined as R17, indicating that hyperglycemia and hypoglycemia with a high rate of change in glucose, glycemic variability, and high glucose fluctuations are applicable. The instability and variability of glucose levels can be verified from SD > 33 and GV ≥ 36 in patients 2, 11, and 12 from Table 5 and in patients 2, 7, 16, and 23 from S1. It was observed that patient 23 had an MODD of 8.5, a CONGA of 0 with SD > 33, and MAGE > 40, depicting higher glycemic variability and high glucose fluctuations with hidden hypoglycemia. Hyperglycemia and hypoglycemia in patients can be analyzed using the hyperglycemic and hypoglycemic trend graph. Hyperglycemia in patient 7 can be visualized in Figure 6B, whereas hypoglycemia can be visualized in Figure 6A. The overall glycemic trends from Figure 6C provide a complete insight into the concentration and spikes of hyperglycemic and hypoglycemic states. Hyperglycemia can be further inferred from the intraday glucose trends to identify the post-meal spikes, peaks, and lows at specific time intervals. The overall glucose trends from “CGM Trace” provide insights into changes and patterns over a specific period.

Normoglycemia can be observed in patients 4, 10, 12, and 15 from Table 4; patients 6, 7, 9, and 13 from Table 5; and patients 4, 10, 13, 21, and 36 from S1. In this case, “This is a normal condition” is displayed on the dashboard.

## 4. Discussion

The current work is the first among the few studies investigating the need and implementation of statistical metrics and their combinational assessment in AGP profiles. The limitations of the AGP profile, i.e., the lack of insightful metrics to identify patients’ response to medication, frequency, reoccurrences, and persistence of glycemic variability, are addressed by “CGM Trace”. “CGM Trace” presents all metrics, trends, and findings for a selected period (from 1 day to n days) unlike the 14 days of the AGP report. Hyperglycemia and hypoglycemia can be accurately identified using the combined assessment of the metrics MG, GV, TAR-I, TAR-II, TIR, and MAGE [15,17,19,24,25,26,28,30,32,33,34,35,40,41,42,43,44,45,46]. The metrics are combined, and the decision rules are formulated. The combined metrics correlate with the intraday trends, the overall glucose trends, and the IQR [15,28,30,32,33,34,35,40,41,42,43]. The GV and IQR correlate with SD, CONGA, and MODD [35,48].

Decision rules and recommendations were combined based on the literature studies in association with different CGM metrics. The decision rules were verified in the clinical study of 15 patients and the public dataset of 52 patients. The metrics, trends, and findings can assist doctors in identifying glycemic control, variability, fluctuations, peaks, and lows that correlate with hidden diseases. The validation of the decision rules was performed by applying the MANOVA test, where *p* < 0.05 is obtained, depicting a strong correlation between the metrics in all rules. With the combined assessment of MG, GV, SDR, TAR-I, TAR-II, TIR, and MAGE, the key findings identified in the current work are as follows: (1.) Hyperglycemia with a high rate of change in glucose, high glycemic variability and high glucose fluctuations can be identified using MG, GV, SDR, TAR-I, TAR-II, TIR, and MAGE; (2.) Hyperglycemia with a high rate of change in glucose can be identified using MG, GV, SDR, TAR-I, TAR-II, TIR, and MAGE; (3.) Hyperglycemia with high glycemic variability can be identified using MG, GV, SDR, TAR-I, TAR-II, TIR, and MAGE; (4.) Hyperglycemia with high glucose fluctuations can be identified using MG, GV, SDR, TAR-I, TAR-II, TIR, and MAGE; (5.) Hypoglycemia with a high rate of change in glucose, high glycemic variability, and high glucose fluctuations can be identified using MG, GV, SDR, TAR-I, TAR-II, TIR, and MAGE; (6.) Hypoglycemia with a high rate of change in glucose can be identified using MG, GV, SDR, TAR-I, TAR-II, TIR, and MAGE; (7.) Hypoglycemia with high glycemic variability can be identified using MG, GV, SDR, TAR-I, TAR-II, TIR, and MAGE; (8.) Hypoglycemia with high glycemic fluctuations can be identified using MG, GV, SDR, TAR-I, TAR-II, TIR, and MAGE; (9.) Even under the presence of normal MG, TIR, SDR, GV, and MAGE, hidden hyperglycemia can be observed; (10.) Even under the presence of normal MG, TIR, SDR, GV, and MAGE, hidden hyperglycemia with a high rate of change in glucose levels can be observed; (11.) Even under the presence of normal MG, TIR, SDR, GV, and MAGE, hidden hyperglycemia with high glycemic variability can be observed; (12.) Even under the presence of normal MG, TIR, SDR, GV, and MAGE, hidden hyperglycemia with high glucose fluctuations can be observed; (13.) Even under the presence of normal MG, TIR, SDR, GV, and MAGE, hidden hypoglycemia can be observed; (14.) Even under the presence of normal MG, TIR, SDR, GV, and MAGE, hidden hypoglycemia with a high rate of change in glucose levels can be observed; (15.) Even under the presence of normal MG, TIR, SDR, GV, and MAGE, hidden hypoglycemia with high glycemic variability can be observed; (16.) Even under the presence of normal MG, TIR, SDR, GV, and MAGE, hidden hypoglycemia with high glucose fluctuations can be observed; (17.) Under the presence of normal MG, TIR, SDR, GV, and MAGE, hyperglycemia and hypoglycemia can be identified; (18.) Under the presence of normal MG, TIR, SDR, GV, and MAGE, hyperglycemia and hypoglycemia with a high rate of change in glucose levels can be identified; (19.) Under the presence of normal MG, TIR, SDR, GV, and MAGE, hyperglycemia and hypoglycemia with high glucose fluctuations can be identified; (20.) Under the presence of normal MG, TIR, SDR, GV, and MAGE, hyperglycemia and hypoglycemia with high glucose fluctuations can be identified.

The glycemic control in patients with diabetes is unpredictable. Many lifestyle factors, such as meal intake, physical activity, and medications, affect glycemic variability. It is crucial to identify the patterns and hidden signs of hyperglycemia and hypoglycemia, through the rate of change in glucose levels, fluctuations, and variability, to reduce the risk of unstable glucose levels and underlying diseases. Understanding the glycemic patterns and variability will help the doctor to make informed decisions regarding treatment changes and adjustments.

This work has shown that the incorporation of the AGP metrics, the added metrics, and the graphical trends provides an insightful representation of a patient’s glycemic levels. This work has also shown that the metrics can be combined and formed as decision rules into a supporting tool, “CGM Trace”, to assist doctors with insightful information, ease of interpretability, and better glycemic management of patients.

“CGM Trace” differs from the AGP report by combining statistical metrics in a single place and providing findings for 20 cases. Each “Finding” defines a clear status of the patient’s glucose levels, which can be accessed at any time. The glucose profile from “CGM Trace” can be saved during every visit and assessed to create a tailored treatment plan, which is lacking in the AGP report. “CGM Trace” enhances the AGP report by providing insights into patients’ complete glucose profile, interday GV, intraday GV, hidden, and temporal patterns of glucose fluctuations using combined metrics and graphical trends for any selected period, which is lacking in the AGP report. The doctor can examine the metrics, trends, and decision rules to create treatment adjustments and strategies in the case of a rise in glucose levels, post-meal spikes, midnight lows, glucose fluctuations, GV patterns, response to medications, and the causes, duration, frequency, severity, and recurrence of hyperglycemia and hypoglycemia. The interpretation of day-to-day or half-day inconsistencies is straightforward in “CGM Trace”. Minute data details below 10% of the lowest glucose levels can be visualized using intraday, hyperglycemic, and hypoglycemic graphs, which are lacking in the AGP report. A report can be generated from “CGM Trace” for maintaining records. “Glucose Profiles” are a rapid method to analyze the glucose trends/graphs ranging from 1 day to n number of days, proving to be a valuable tool during the emergency condition of the patient. Response to medication, persistence of glycemic peaks or lows, and reoccurrences of hyperglycemia and hypoglycemia can be analyzed visually using the “Glucose profiles” of “CGM Trace”, which is lacking in the AGP report. This work showcases the potential of a combinational assessment of metrics and respective findings as a decision-support tool to enhance the AGP report. The findings highlight the potential of identifying many combinational cases of hyperglycemia and hypoglycemia, which indicate underlying diseases. The management of glycemic variability, fluctuations, and hidden patterns improves overall glucose control for a better lifestyle and diabetic management using “CGM Trace”.

Despite these findings, there are limitations in the proposed work. The proposed work was implemented in a dataset of 67 patients. It is important to acknowledge that the results may therefore not be directly transferable to datasets including patients who met the exclusion criteria. Diabetes coexists with many chronic conditions, such as chronic kidney disease, cardiovascular disease, and neurological conditions. In this work, the correlation between metrics was not identified, as the data originated from different sources where the patient demographics and medical history were different, resulting in a lack of homogeneity. The proposed work can be further enhanced by considering a large number of datasets with coexisting conditions and homogeneity to find correlation between the variables. The effect of chronic conditions on the variables among the patient demographics will be taken as future work. The proposed “CGM Trace” is not a real-time monitoring technique, as the data from the CGM device have to be uploaded manually by the user. Another limitation of the proposed work is that the user must have an Internet connection to access the webpage. As a future work, the webpage will be developed into a mobile application (app) for ease of access and deployed on a global server. The user-friendly interface of the app will be crafted to avail maximum benefit from the app to people in rural and urban settings. The design will consider literacy levels, language preferences, and device compatibility at the front-end. The accessibility of important features in offline mode and training programs, by collaborating with local healthcare workers in rural areas to provide education about the app, will be considered for the accessibility of the technology. The findings during every doctor visit for each patient can be gathered and integrated with machine learning algorithms for prediction and treatment recommendations or better diabetic management. These findings can help future researchers to develop optimal glycemic control strategies. The efficacy of new drugs being assessed by identifying their response from the findings can be considered as future work.

In summary, the findings of the proposed work suggest the enhancement of the AGP report in identifying glucose variability with a combinational assessment of metrics at any given time. However, the enhancement of the AGP profile is an attempt to provide a comprehensive view of glucose control and improvement in the area of diabetes management.

## 5. Conclusions

The core competencies of the current work are the findings from the decision rules, which were built upon metrics. The findings and metrics were statistically validated and implemented on a dataset of 67 patients. “CGM Trace” is a decision-assistive supporting tool that presents summarized findings on patients’ glucose profiles with metrics and graphical representations, thus enhancing the AGP report. The research gap identified in the literature, i.e., the minimum days of data requirement and the ease of interpretability in the AGP report, was overcome by incorporating statistical metrics, along with the easy-to-interpret graphical illustrations, which can be accessed at any time through “CGM Trace”. Apart from the significance of the current study to clinicians, the software is also an excellent tool for personal care, making informed decisions, controlling glucose management, and ultimately improving long-term health outcomes in diabetic management.

## Figures and Tables

**Figure 1 diagnostics-14-00436-f001:**
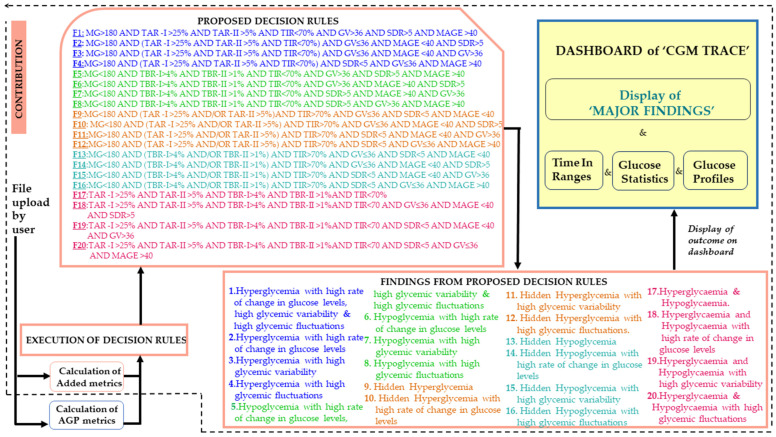
Decision assistive system from “CGM Trace”.

**Figure 2 diagnostics-14-00436-f002:**
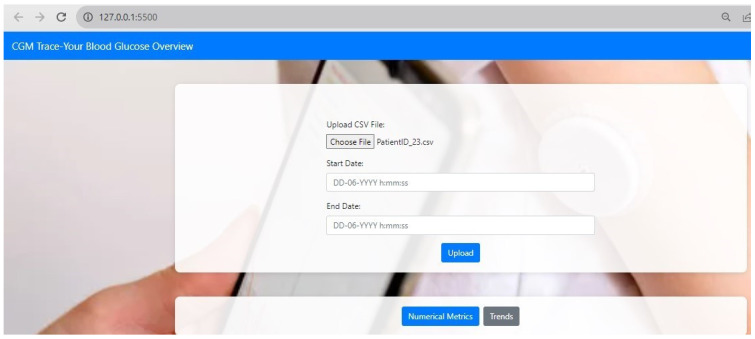
Frontpage of the web interface.

**Figure 3 diagnostics-14-00436-f003:**
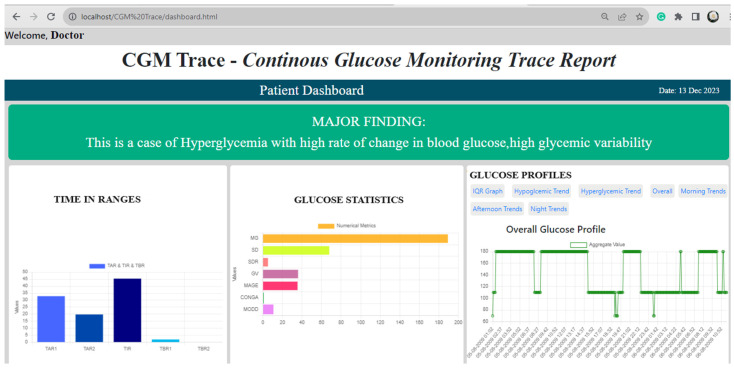
The findings page of “CGM Trace”, accessible to the doctor.

**Figure 4 diagnostics-14-00436-f004:**
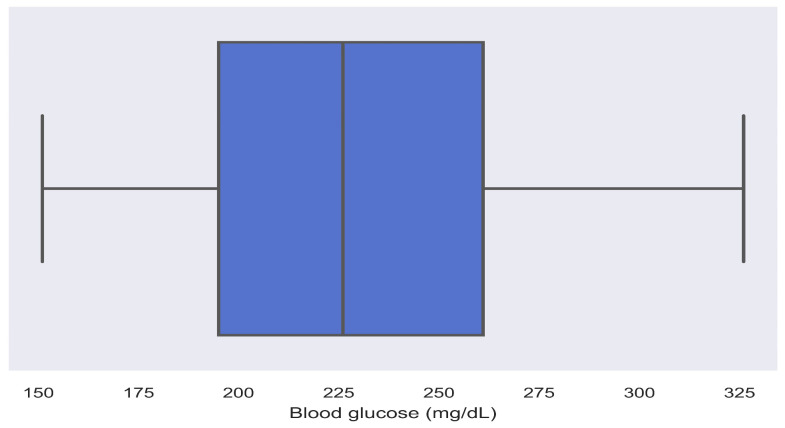
IQR representing glucose distribution in 24 h.

**Figure 5 diagnostics-14-00436-f005:**
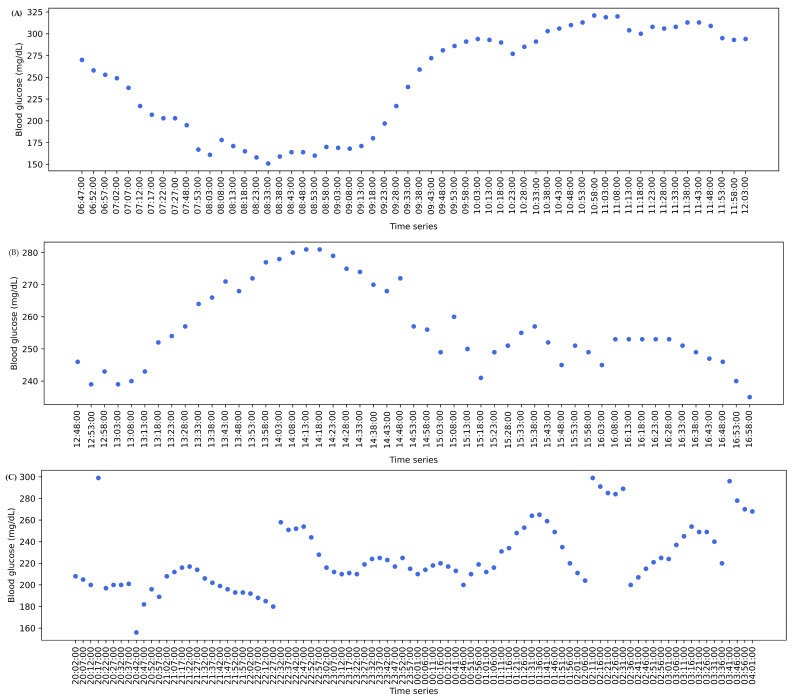
Graphical illustration of glycemic variability in a patient. (**A**) Intraday glucose trends in the morning. (**B**) Intraday glucose trends in the afternoon. (**C**) Intraday glucose trends in the night.

**Figure 6 diagnostics-14-00436-f006:**
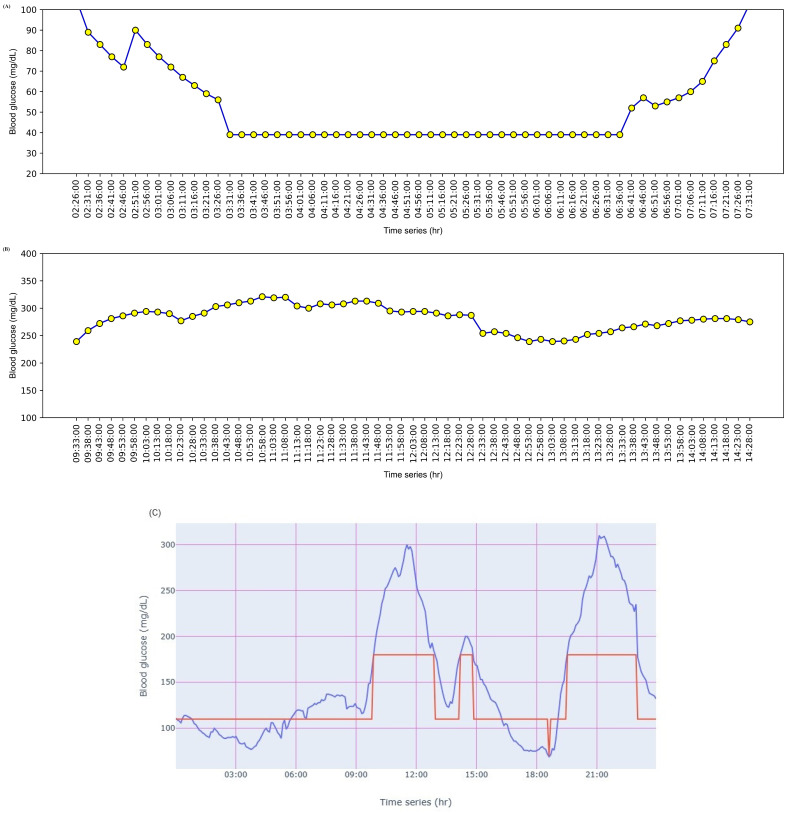
Graphical illustration of glycemic variability in a patient. (**A**) Hypoglycemic trend. (**B**) Hyperglycemic trend. (**C**) Overall glycemic trends.

**Table 1 diagnostics-14-00436-t001:** Patient demographics from the clinical study and JCHR-JAEB dataset (n = 67).

Age (in groups) (1–30): (30–60): (60–90):	12 to 65 years M: 29, F: 29 M: 4, F: 3 M: 1, F: 1
Gender	M = 34, F = 33
Diabetes duration	At least 1 year of clinical diagnosis with diabetes
HbA1C	Measured between 5.0 and 10.5
Inclusion criteria	Patients on CGM sensor with proper mental health and cognition
Exclusion criteria	Pregnant and lactating women, patients with diabetic ketoacidosis, seizure disorder, active infection, muscular condition, cancer patients, and cystic fibrosis

Abbreviations: M, male; F, female; HbA1C, glycated hemoglobin; CGM, continuous glucose monitoring.

**Table 2 diagnostics-14-00436-t002:** Decision rules from the combined assessment of metrics programmed in “CGM Trace”.

R.No	Decision Rule for Different Findings
R1	If **MG > 180** AND **TAR-I > 25%** AND **TAR-II > 5%** AND **TIR < 70%** AND **GV > 36** AND **SDR > 5** AND **MAGE > 40** Then This is a case of hyperglycemia with a high rate of change in glucose levels, high glycemic variability, and high glucose fluctuations
R2	If **MG > 180** AND **(TAR-I > 25%** AND **TAR-II > 5%** AND **TIR < 70%)** AND GV ≤ 36 AND MAGE < 40 AND **SDR > 5** Then This is a case of hyperglycemia with a high rate of change in glucose
R3	If **MG > 180** AND **(TAR-I > 25%** AND **TAR-II > 5%** AND **TIR < 70%)** AND SDR < 5 AND MAGE < 40 AND **GV > 36** Then This is a case of hyperglycemia with a high glycemic variability
R4	If **MG > 180** AND **(TAR-I > 25%** AND **TAR-II >5%** AND **TIR < 70%)** AND SDR < 5 AND GV ≤ 36 AND **MAGE > 40** Then This is a case of hyperglycemia with high glucose fluctuations
R5	If **MG < 180** AND **TBR-I > 4%** AND **TBR-II > 1%** AND **TIR < 70%** AND GV > 36 AND SDR > 5 AND **MAGE > 40** Then This is a case of hypoglycemia with a high rate of change in glucose levels, high glycemic variability, and high glucose fluctuations
R6	If **MG < 180** AND **TBR-I > 4%** AND **TBR-II > 1%** AND **TIR < 70%** AND GV ≤ 36 AND MAGE < 40 AND **SDR > 5** Then This is a case of hypoglycemia with a high rate of change in glucose
R7	If **MG < 180** AND **TBR-I > 4%** AND **TBR-II > 1%** AND **TIR < 70%** AND SDR < 5 AND MAGE <40 AND **GV > 36** Then This is a case of hypoglycemia with high glycemic variability
R8	If **MG < 180** AND **TBR-I > 4%** AND **TBR-II > 1%** AND **TIR < 70%** AND SDR < 5 AND GV ≤ 36 AND **MAGE > 40** Then This is a case of hypoglycemia with high glucose fluctuations
R9	If MG < 180 AND **(TAR-I > 25% OR TAR-II >5%)** AND TIR > 70% AND GV ≤ 36 AND SDR < 5 AND MAGE <40 Then This is a case of hidden hyperglycemia
R10	If MG < 180 AND **(TAR-I > 25% OR TAR-II > 5%)** AND TIR > 70% AND GV ≤ 36 AND MAGE < 40 AND **SDR > 5** Then This is a case of hidden hyperglycemia with a high rate of change in glucose levels
R11	If MG > 180 AND **(TAR-I > 25% OR TAR-II > 5%)** AND TIR > 70% AND SDR < 5 AND MAGE < 40 AND **GV > 36** Then This is a case of hidden hyperglycemia with high glycemic variability
R12	If MG > 180 AND **(TAR-I > 25% OR TAR-II > 5%)** AND TIR > 70% AND SDR < 5 AND GV ≤ 36 AND **MAGE > 40** Then This is a case of hidden hyperglycemia with high glucose fluctuations
R13	If MG < 180 AND **(TBR-I > 4% OR TBR-II > 1%)** AND TIR > 70% AND GV ≤ 36 AND SDR < 5 AND MAGE < 40 Then This is a case of hidden hypoglycemia
R14	If MG < 180 AND **(TBR-I > 4% OR TBR-II > 1%)** AND TIR > 70% AND GV ≤ 36 AND MAGE <40 AND **SDR > 5** Then This is a case of hidden hypoglycemia with a high rate of change in glucose levels
R15	If MG < 180 AND **(TBR-I > 4% OR TBR-II > 1%)** AND TIR > 70% AND SDR < 5 AND MAGE <40 AND **GV > 36** Then This is a case of hidden hypoglycemia with high glycemic variability
R16	If MG < 180 AND **(TBR-I > 4% OR TBR-II > 1%)** AND TIR > 70% AND SDR < 5 AND GV ≤ 36 AND **MAGE > 40** Then This is a case of hidden hypoglycemia with high glucose fluctuations
R17	If **(TAR-I > 25% OR TAR-II >5%)** AND **(TBR-I > 4% OR TBR-II > 1%)** AND **TIR < 70%** Then This is a case of hyperglycemia and hypoglycemia at different intervals
R18	If **(TAR-I > 25%** AND **TAR-II > 5%)** AND **(TBR-I > 4%** OR **TBR-II > 1%)** AND **TIR < 70** AND GV ≤ 36 AND MAGE <40 AND **SDR > 5** Then This is a case of hyperglycemia and hypoglycemia at different intervals with a high rate of change in glucose levels
R19	If **(TAR -I > 25%** OR **TAR-II > 5%)** AND **(TBR-I > 4%** AND **TBR-II > 1%)** AND **TIR < 70** AND SDR < 5 AND MAGE <40 AND **GV > 36** Then This is a case of hyperglycemia and hypoglycemia at different intervals with high glycemic variability
R20	If **(TAR-I > 25%** OR **TAR-II > 5%)** AND **(TBR-I > 4%** AND **TBR-II > 1%)** AND **TIR < 70** AND SDR < 5 AND GV ≤ 36 AND **MAGE > 40** Then This is a case of hyperglycemia and hypoglycemia at different intervals with high glucose fluctuations

Abbreviations: R.No, Rule number; MG, mean glucose; GV, glucose variability; TIR, time in range; TAR, time above range; TBR, time below range; SDR, standard deviation rate of change; MAGE, mean amplitude of glycemic excursions.

**Table 3 diagnostics-14-00436-t003:** Assumptions and *p*-value from the MANOVA one-way test.

No.	Assumptions	Test Performed	Transformation Performed	*p*-Value
1	Normality distribution	Normality test	Inverse DF	-
2	Homogeneity of covariance matrices	Box’s M test	-	0.531
3	Effect of independent variables on dependent variables	Tests of Between-Subjects Effects	-	0.791
**4**	**Overall significance**	**Wilk’s Lambda**	**-**	**0.001**

Abbreviations: No, number; DF, distribution function.

**Table 4 diagnostics-14-00436-t004:** CGM data analyzation from “CGM Trace”—Clinical study outcome.

P				Metrics								
	SDR	MG	SD	GV (%)	TIR (%)	TAR-I (%)	TAR-II (%)	TBR-I (%)	TBR-II (%)	MODD	CONGA	MAGE
1	5.4	96.6	42.2	34.9	65.3	3.07	0	11.5	18.4	7.4	0	73.5
2	5.5	117.7	48.9	31.1	64.4	22.2	5	5.9	1.7	7.7	2.6	96.2
3	4.0	126.1	36.1	29.4	86.1	8.8	0	5.5	0	6.7	8.8 × 10^−16^	44.9
4	3.94	154.4	33.0	36	76.5	23.4	0	0	0	4.2	1.0	19.7
5	6.0	96.0	27.3	37.4	68.6	0	0	18.7	12.1	4.7	1.7 × 10^−15^	51.5
6	5.86	90.2	38.7	31.4	59.4	2.7	0	17.2	20.5	5.6	0	37.5
7	3.5	74.1	31.2	45.3	47.3	0	0	14.0	38.5	4.2	8.8 × 10^−16^	38
8	4.0	152.3	32.5	30	73.6	21.0	5.0	0	0	4.7	4.2 × 10^−15^	39.5
9	4.5	151.6	32.5	25.2	75	23.4	0	1.5	0	3.4	0	17.4
10	2.7	126.8	29.2	23	96.6	1.6	0	1.6	0	2.1	8.8 × 10^−16^	30.5
11	5.6	180.4	38.8	36.4	65	27.7	5.2	0	0	5.4	0	46.7
12	3.6	133.4	34.9	26.8	84.4	7.7	0	7.7	0	4.8	0	44.1
13	6	186.7	41	44.2	61.5	31.4	7	0	0	5.3	0	42.9
14	5.0	190.9	47.2	39.3	60.4	26.0	6	5	3	4.2	1.7 × 10^−15^	33.2
15	2.97	125.2	22.3	16.2	100	0	0	0	0	2.6	4.4 × 10^−16^	36

Abbreviations: P, patient number; SDR, standard deviation rate of change; SD, standard deviation; MG, mean glucose; GV, glucose variability; TIR, time in range; TAR, time above range; TBR, time below range; MODD, mean of daily differences; CONGA, continuous overall net glycemic action; MAGE, mean amplitude of glycemic excursions.

**Table 5 diagnostics-14-00436-t005:** CGM data analysis from “CGM Trace”—Public dataset from JCHR-JAEB.

P				Metrics								
	SDR	MG	SD	GV (%)	TIR (%)	TAR-I (%)	TAR-II (%)	TBR-I (%)	TBR-II (%)	MODD	CONGA	MAGE
1	5.5	176.9	52.6	29.6	50	39.5	10.4	0	0	2.7	4.4 × 10^−16^	26.6
2	3.1	172.9	103	53.6	48	19.5	28.7	4.6	0	4.1	5.9 × 10^−16^	28.5
3.	2.5	164.7	48.1	29.2	65.5	26.8	4.6	0	3	3.7	8.4 × 10^−16^	43.1
4	4.3	140.6	30	32.3	84.2	9.4	4.6	14.1	1.6	4.1	8.8 × 10^−16^	42.6
5	5.3	151	22	33	69.1	26.0	4.3	0.2	0	4.2	8.8 × 10^−16^	36.7
6	3	128.5	40.7	31.7	87.5	12.4	0	0	0	4.2	0	27.2
7	2.6	154.4	30.9	20	84.0	15.9	0	0	0	3.5	4.4 × 10^−16^	40
8	31.1	193.9	60.4	31.1	44.9	37.4	17.3	0.2	0	4.9	1.70 × 10^−15^	61.5
9	2.1	141	38.6	27.3	83.7	15.8	0	0	0	4.1	0	37
10	1.7	166.7	49.7	29.7	73.8	14.9	10.0	0	0	2.4	0	66.6
11	3.9	163.6	79	48.2	47.4	24.6	15	6.4	5.9	5.5	7.8 × 10^−16^	34.8
12	4.3	160	79.9	49.8	44.8	29.3	7.7	3.4	13.8	5.1	5.8 × 10^−16^	130.2
13	4.6	155.3	30.5	26	74	23.7	2.8	2.6	0	5.5	0	37.8
14	4.4	122	86.7	38.9	40.9	25.7	33.3	0	0.1	5	8.8 × 10^−16^	42
15	7.0	218.0	53.6	24.6	26.8	73.1	0	0	0	9.2	1.4 × 10^−16^	40.7

Abbreviations: P, patient number; SDR, standard deviation rate of change; SD, standard deviation; MG, mean glucose; GV, glucose variability; TIR, time in range; TAR, time above range; TBR, time below range; MODD, mean of daily differences; CONGA, continuous overall net glycemic action; MAGE, mean amplitude of glycemic excursions.

## Data Availability

The datasets used and/or analyzed in this study are available from the corresponding author upon reasonable request and on the website of the JCHR-JAEB center for Health Research.

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
