# Peer review of "Enhancement of Ambulatory Glucose Profile for Decision Assistance and Treatment Adjustments"

_diagnostics, 2024, doi:10.3390/diagnostics14040436_

Round 1

Reviewer 1 Report (New Reviewer)

Comments and Suggestions for Authors

The AGP lacks comprehensive data analytics for the clinicians and physicians to make an informed decision. To overcome/bridge the incomplete information, the authors proposed  to overcome the challenges of missing insightful information, difficulty in graphical interpretation, and creating a platform for decision assistance. Eventually this will help patients with hyper and hypoglycemic conditions. Statistical analysis was sufficiently performed. The research gaps identified by the authors are well described and hence, the need to address this shortcoming.

  1. The authors identified the number of datasets and the variability as one of the limitations of the study. The manuscript in its present form significantly lacks comprehensive patient statistics such as age range, gender, etc. A table needs to be included stratifying the 67 patients.
  2. Did the patient data originate from a single place or was well scattered. Their diet, exercise, etc play a pivotal role in regulating blood glucose and can be a significant factor depicting results. 
  3. Feasibility of the technology in rural (not so well to do) communities is an issue. How feasible is this technology according to the authors especially since they are looking to convert it into an App-based technology. 
  4. Also, based on the analyzed patterns, do the authors draw any significant correlations with respect to variables addressed in this technology. Whether a selected few variables play a major role and are sufficient to draw inferences or each variable is important?
  5. Is it a real time monitoring technique or the data is available in packets and at certain intervals. What is the data processing time or time between which, the data is analyzed and then available to the clinician?
  6. Lastly, I see only 34 references cited. However, the manuscript shows 51 references.

Author Response

Original Manuscript ID: 2692617

Original Article Title: " Enhancement of AGP Profile for Decision Assistance and Treatment Titrations."

To: Diagnostics Editor

Re: Response to reviewers

Dear Editor,

Thank you for allowing a resubmission of our manuscript, with an opportunity to address the reviewers’ comments.

Grammar has been checked from Grammarly enterprise edition software. Major changes we have made in the manuscript are (a) Adding a detailed explanation of findings from (F1-20) (b) Updated the MANOVA table and explaining the process of carrying out the MANOVA one-way test (c) Adding and updating the references as suggested (d) Added future work as suggested by the reviewers

We are uploading (a) Our point-by-point response to the comments (below) (response to reviewers), (b) An updated manuscript with yellow highlighting indicating changes (Highlighted copy), (c) A clean updated manuscript without highlights (Main Manuscript), (d) Plagiarism report and (e) Supplementary material.

Best regards,

P.Vijayakumar.

Reviewer 1

  1. The authors identified the number of datasets and the variability as one of the limitations of the study. The manuscript in its present form significantly lacks comprehensive patient statistics such as age range, gender, etc. A table needs to be included stratifying the 67 patients.

Authors Response: The authors apologize for missing out on the statistics of the patient. The authors have added the patient’s demographics in the manuscript.

Authors Action: The authors have updated the manuscript by adding Table 1 in page number 3 (line 134) under 2.1 CGM data collection as,

Table 1 Patient’s Demographics from Clinical Study and JCHR-JAEB dataset (n=67)

Age (in groups)

(1-30):

(30-60):

(60-90):

12 to 65 years

M:29 ,F:29

M: 4,F:3

M: 1,F:1

Gender

M=34,F=33

Diabetes duration

At least 1 year of clinical diagnosis with diabetes

HbA1C

Measured between 5.0 to 10.5

Inclusion criteria

Patients on CGM sensor with proper mental health and cognition.

Exclusion criteria

Pregnant and lactating women, patients with diabetic ketoacidosis, seizure disorder, active infection, muscular condition, cancer patients, and cystic fibrosis

Abbreviations: M, male;F,female;HbA1C, glycated hemoglobin; CGM, continuous glucose monitoring

2.Did the patient data originate from a single place or was well scattered. Their diet, exercise, etc play a pivotal role in regulating blood glucose and can be a significant factor depicting results.

Authors Response: The patients are thankful to the esteemed reviewers for their remark on origin of patient data and significant factors depicting results.

(a) The patient data from the clinical study originate from Tamil Nadu, India and the patient data considered from the public dataset is originated from USA, France, Italy, and Israel collected by JCHR JAEB.

(b) The authors agree that the diet, exercise play a pivotal role in regulating blood glucose which has a significant effect on the results. The authors want to state that the ‘CGM Trace’ is a universal software which provides a comprehensive report on glucose levels by implementing standardized metrics. The effect of meals, and physical activity will reflect at the glucose levels which is thoroughly analysed from ‘CGM Trace’. Therefore, CGM Trace is a suitable tool to provide individualized reports based on patients’ data and origin.

Authors Action: The authors have updated the manuscript by,

(a) adding the origin of data in page number 3 (line 129) under 2.1 CGM data collection as,

In this work, clinical study data originate from Tamil Nadu, India and USA, France, Italy, and Israel collected by JCHR JAEB.

(b) adding that ‘CGM Trace’ is a universal software developed from accommodating meals, physical activity in page number 5 (line 168) under 2.2 Metrics Incorporated in ‘CGM Trace’ as,

The proposed ‘CGM Trace’ is a universal software developed by implementing standardized metrics. Irrespective of origin of data, meals, and physical activity will reflect at the glucose levels which is thoroughly analyzed from ‘CGM Trace’.

3.Feasibility of the technology in rural (not so well to do) communities is an issue. How feasible is this technology according to the authors especially since they are looking to convert it into an App-based technology.

Authors Response: The authors would like to thank the esteemed reviewers for their remark on feasibility of the technology in rural areas. The authors want to state that user friendly interface of the App will be designed based on literacy level, language preferences and device compatibility. Accessibility of important features in offline mode, training programs for educating people about the App and follow -up are planned to consider for everyone to benefit from the App.

Authors Action: The authors have updated the manuscript by adding the reviewers comment in future work in page number 19 (line 682) under 4. Discussion as,

User friendly interface of the app will be crafted for availing maximum benefit from the app by rural and urban people. The design will consider the literacy levels, language preferences and device compatibility at the front-end. Accessibility of important features in offline mode and training programs by collaborating with local healthcare workers in rural areas for educating about the app will be considered for feasibility of technology.

4.Also, based on the analyzed patterns, do the authors draw any significant correlations with respect to variables addressed in this technology. Whether selected few variables play a major role and are sufficient to draw inferences or each variable is important?

Authors Response: The authors are thankful to the esteemed reviewers for their comment on drawing any significant correlation between the variables. The authors want to state that as the proposed work was designed to analyze patterns and trends from the glucose levels, all diabetic patients were considered. However, these patients have chronic diseases such as cardiovascular disease, chronic kidney disease, and neurological disorders. To find a correlation between the parameters, homogeneity in the demographics of patients must be considered with similar age group, same diseases and a control group. As the query looks interesting and a potential problem statement, the authors would consider it as a future work to find the correlation between the variables and if the correlation vary w.r.t age, gender and if the effect of variables differ with different coexisting diseases along with diabetes.

Authors Action: The authors have updated the manuscript by adding about correlation between variables as limitation and future work in page number 19 (line 670 to 677) under 4. Discussion as,

Diabetes coexists with many chronic conditions such as chronic kidney disease, cardiovascular disease, and neurological conditions. In this work, the correlation between metrics is not performed as data originates from different sources where demographics and patient's medical history are different lacking homogeneity. The proposed work can be further enhanced by considering a large number of datasets with coexisting conditions and homogeneity in data for finding correlations between the variables. The effect of chronic conditions on the variables among patient’s demographics will be taken as future work.

5.Is it a real time monitoring technique or the data is available in packets and at certain intervals. What is the data processing time or time between which, the data is analyzed and then available to the clinician?

Authors Response: The authors are thankful to the esteemed reviewers for their comment on real time monitoring technique or if data is available in packets at certain intervals.

                                    The authors want to state that ‘CGM Trace’ is not a real time monitoring technique.

Authors Action: The authors have updated the manuscript by adding about monitoring technique as limitation in page number 19 (line 677) under 4. Discussion as,

The proposed ‘CGM Trace’ is not a real-time monitoring technique as the data from CGM device has to be uploaded manually by the user.

6.Lastly, I see only 34 references cited. However, the manuscript shows 51 references.

Authors Response: The authors apologize for the inconvenience caused due to the wrong citation of references. The authors have updated the references in the manuscript.

Authors Action: The authors have updated the references in the manuscript and highlighted them in yellow. The updated references are provided in page number 21.

Reviewer 2 Report (New Reviewer)

Comments and Suggestions for Authors

Strange opening statement - best to define AGP in the text as people might not look for the definition in the abstract; additionally, besides mentioning disadvantage in the second paragraph, it helps to motivate this research with the advantages. Plus, at some moment it would help to state what disadvantages you are talking about. 

It is confusing that you use  "fluctuations" and "variability"  interchangeably throughout the text. There are some differences among the two. 

The way that GMI is introduced, it might not be clear what the metric is. 

When you first mention MAGE (line 50), there is no explanation for general audience to appreciate its significance. 

How do you define hidden hypo- and hyperglycemia?

There are fairly commonly used datasets by JAEB, is there a special reason for using those sets reported in 2011?

The introduction of SDR (line 162) is coming from a web site.. can you use instead papers and books that are more persistent (web sites change)?

In line 194: The suggestion of clinician might also include taking a "correction dose of insulin" after taking meal. Moreover, the term "doctor" refers to which type of clinician? Nutritionist or Endocrinologist? Or a board made up both?

In line 200, "it" refers to CONGA or threshold?

Some decision rules appear to overlap or have very subtle differences (e.g., F1 vs F18, F13 vs F14), A more detailed explanation of how each rule distinctly contributes to patient assessment would be beneficial.

Rule F17 seems to oversimplify complex medical scenarios by covering both hyper and hypoglycemia.

It is confusing when you write AND/OR without explanation: OR subsumes AND, for one. 

In rule F17, scientific supports required for this assumption (hyper and Hypoglycemia.   

 at the same time). Authors also have similar claim in line 432. The publication referred below ssupoports coexistance of "rebound" hypo- and hyper-glycemia. However, it doesnt support "concurrent" occurrence of those phenomenon,. In other words, a hypoglycemia can be followed by hyperglycemia (or vice versa), but they are not happening simultaneously.

Hansen KW, Bibby BM. Rebound Hypoglycemia and Hyperglycemia in Type 1 Diabetes. J Diabetes Sci Technol. 2023 May 3:19322968231168379. doi: 10.1177/19322968231168379. This assumption may lead to false results.

Line 263: It is worth to mention link of code repo/webpage, if this project/page is publicly available.

Table 4: manova typically provides a single  p-value related to the overall test, not individual p-values for each combination of condition and metric. 

Line 296: Which software used to run this tests?  readers could also benefit from normality check of data, or if any correction conducted whit MANOVA.

It would be helpful to describe how each label (F1-20) assigned to each patient, before conducting MANOVA. Was this based on the opinion of a clinician?

Line 307: Which software used to run this tests?  readers could also benefit from normality check of data, or if any correction conducted whit MANOVA.

It would be helpful to describe how each label (F1-20) assigned to each patient, before conducting MANOVA. Was this based on the opinion of a clinician?

Line 314: You can simply name tables in supplementary material as S!, S2, and avoid repeating "supplementary material" and confusing reader with Tables inside main text.

Line 325 and later: "Finiding N from Table 1 .... is applicable as a decision rule" repetition could decrease readability of the section. This sentence has been repeatedly used for each finding.

Line 328-329: Is the combination of decision rules into a single  finding is  based on another "higher level" rule? When system is allowed to aggregate findings? A description of this process in methods and material section would be helpful. 

Line 432: those statements could help earlier, plus it helps to use a reference, e.g.: Hansen KW, Bibby BM. Rebound Hypoglycemia and Hyperglycemia in Type 1 Diabetes. J Diabetes Sci Technol. 2023 May 3:19322968231168379. doi: 10.1177/19322968231168379. 

Line 549: GP is generally made to present overview of changes. Then clinician could go back to daily "mean glucose" chart to check variability of  glucose levels. I recommend to elaborate more over the difference of your tool versus other similar tools. 

Line 553: Why access to :"glucose profiles" should be limited to only emergency cases?

Line 56-562: Repetition of exclusion criteria

Future work pointers are weak, how these findings can help further studies?

Comments on the Quality of English Language

An overall and comprehensive proof-reading is needed to enhance typos and grammar errors. Some examples are mentioned above, 

Author Response

Reviewer 2

1.Strange opening statement - best to define AGP in the text as people might not look for the definition in the abstract; additionally, besides mentioning disadvantage in the second paragraph, it helps to motivate this research with the advantages. Plus, at some moment it would help to state what disadvantages you are talking about.

Authors Response: The authors apologize in missing out to define AGP in the opening statement, advantages, disadvantages and motivation statement. The authors have updated the manuscript by defining AGP, and mentioning about the advantage and disadvantages that are leading towards the motivation to carry current work.

Authors Action: The authors have updated the manuscript by

(a) adding the definition in page number 1 (line 30) under 1. Introduction as,

AGP is a comprehensive report of summarized glycemic control in statistical and graphical representations generated from continuous glucose monitoring (CGM) data.

(b) adding the advantage in page number 1 (line 36) under 1. Introduction as,

The statistics, graphs, and predefined glucose target ranges of the AGP report help doctors to make informed decisions and personalized guidance to patients for better diabetic management.

(c) adding the disadvantage and motivation in page number 2, after research gap under 1. Introduction (line 83) as,

Moreover, contextual information i.e., response to medication changes, the effect of ill-ness, meal plans, and physical activity cannot be analyzed from a summarized 14-day report which is a disadvantage. To address these issues, the development of decision support tools and their management to assist users are briefly recommended for the maximum usage of CGM and an attempt to enhance the AGP report [11-12]. This is our motivation to develop ‘CGM Trace’ aiming for better diabetic management.

2.It is confusing that you use "fluctuations" and "variability" interchangeably throughout the text. There are some differences among the two.

Authors Response: The authors apologize for wrongly writing the "fluctuations" instead of "variability”. The authors have updated the manuscript by correcting the words as suggested.

Authors Action: The authors have updated the manuscript by changing "fluctuations" instead of "variability” and highlighted in the manuscript.

3.The way that GMI is introduced, it might not be clear what the metric is.

Authors Response: The authors apologize for missing out to introduce GMI. The authors have provided the introduction to GMI.

Authors Action: The authors have updated the manuscript by introducing GMI in page number 2 (line 45) under 1. Introduction as,

Glucose Management Indicator (GMI) is a measure of approximate HbA1C based on an average glucose level of 14 days.

4.When you first mention MAGE (line 50), there is no explanation for general audience to appreciate its significance.

Authors Response: The authors apologize for missing out the explanation on MAGE. The authors have updated the manuscript by providing the explanation of MAGE.

Authors Action: The authors have updated the manuscript by providing an explanation on MAGE in page number 2 (line 56) under 1. Introduction as,

MAGE is a numerical measure of an average amplitude of upward and downward glucose excursions.

  1. How do you define hidden hypo- and hyperglycaemia?

Authors Response: The authors apologize for missing out to define hidden hyperglycaemia and hidden hypoglycaemia. The authors have updated the manuscript by providing the definition of hidden hyperglycaemia and hidden hypoglycaemia

Authors Action: The authors have updated the manuscript by defining hidden hyperglycaemia and hidden hypoglycaemia in page number 7 (line 271 to line 276) under 2.3 Proposed Decision Rules as,

Hidden hyperglycemia is a condition when either or both of TAR-I>25%,TAR-II>5% are above target levels even when the overall glucose levels are in target range (MG<180,TIR>70%, GV≤36, SDR<5 ,MAGE <40).Similarly, hidden hypoglycemia is a condition when either or both of TBR-I>4%,TBR-II>1% are above target levels even when the overall glucose levels are in target range (MG<180,TIR>70%, GV≤36, SDR<5 ,MAGE <40).  

6.There are fairly commonly used datasets by JAEB, is there a special reason for using those sets reported in 2011?

Authors Response: The authors are thankful to the esteemed reviewers for their remark on reason for using ‘closed-loop control to range system’.

Authors Action: The authors want to state that ‘closed-loop control to range system’ dataset had the details related glucose values at specific date and time from the sensor, date and time of CGM insertion and removal, along with the patient’s historical condition in details which was lacking in other datasets.

7.The introduction of SDR (line 162) is coming from a web site.. can you use instead papers and books that are more persistent (web sites change)?

Authors Response: The authors are thankful to the esteemed reviewers for their suggestion to use supporting reference from research paper as websites may change. The authors have added the reference for SDR from research paper as suggested.

Authors Action: The authors have updated the reference [17] in the manuscript as,

Clarke W, Kovatchev B. Statistical tools to analyze continuous glucose monitor data. Diabetes Technol Ther. 2009;11(S1) :S-45-S-54. doi:10.1089/dia.2008.0138

8.In line 194: The suggestion of clinician might also include taking a "correction dose of insulin" after taking meal. Moreover, the term "doctor" refers to which type of clinician? Nutritionist or Endocrinologist? Or a board made up both?

Authors Response: The authors are thankful to the esteemed reviewer for their suggestion of "correction dose of insulin" and if doctor refers to nutritionist or endocrinologist. The authors have updated the manuscript by including the suggestions.

Authors Action: The authors have updated the manuscript by

(a) adding the suggestion from the reviewer in page number 5 (line 215) under 2.2.1 Added Metrics

as,

correction dose of insulin

(b) adding the reference of doctor as ‘endocrinologist’ in page number 1 (line 19) under Abstract as,

However, the doctor (endocrinologist)

9.In line 200, "it" refers to CONGA or threshold?

Authors Response: The authors apologize for the confusion created due to “it” referring to CONGA or threshold. The authors have corrected the word as suggested in the manuscript.

Authors Action: The authors have updated manuscript in page number 6 (line 242) under 2.2.1 Added Metrics as,

However, it was observed that CONGA increases gradually

10.Some decision rules appear to overlap or have very subtle differences (e.g., F1 vs F18, F13 vs F14), A more detailed explanation of how each rule distinctly contributes to patient assessment would be beneficial.

Authors Response: The authors apologize for missing out to provide a detail explanation about the decision rules. The authors have incorporated a detailed explanation in the manuscript by,

(a) replacing the F1-F20 as R1-R20 (rule 1-rule 20) to avoid confusion.

(b) distinct rules are formed to emerge as a single outcome from 20 decision rules and provided the explanation for each.

Authors Action: The authors have updated the manuscript by providing an explanation on each rule in page number 6 (line 268) under 2.3 Proposed Decision Rules as,

Decision rules are formed from combined literature and clinical studies categorized based on poor and unstable glycemic control as presented in Table 2. The decision rules are categorized into hyperglycemia, hypoglycemia, hidden hyperglycemia, hidden hypoglycemia, hyperglycemia, and hypoglycemia. Hidden hyperglycemia is a condition when either or both of TAR-I>25%,TAR-II>5% are above target levels even when the overall glucose levels are in target range (MG<180,TIR>70%, GV≤36, SDR<5 ,MAGE <40).Similarly, hidden hypoglycemia is a condition when either or both of TBR-I>4%,TBR-II>1% are above target levels even when the overall glucose levels are in target range (MG<180,TIR>70%, GV≤36, SDR<5 ,MAGE <40). 

R1 is applicable to the patient when MG, TAR-I, TAR-II, GV, SDR and MAGE are above the target levels and TIR is below the target level indicating Hyperglycemia.R2 is applicable to the patient when MG, TAR-I,TAR-II,SDR are above the target levels ,TIR is below the target level, GV and MAGE are in target level indicating Hyperglycemia with high rate of change in glucose. R3 is applicable to the patient when MG, TAR-I, TAR-II, GV are above the target levels, TIR is below the target level, SDR and MAGE are in target level indicating Hyper-glycemia with high glycemic variability. R4 is applicable to the patient when MG, TAR-I, TAR-II , MAGE are above the target levels, TIR is below the target level, SDR and GV are in target level indicating Hyperglycemia with high glucose fluctuations.

R5 is applicable to the patient when TBR-I, TBR-II, GV, SDR, MAGE are above the target levels and MG, TIR are below the target levels, indicating Hypoglycemia.R6 is ap-plicable to the patient when TBR-I, TBR-II, SDR are above the target levels ,MG,TIR are be-low the target levels, GV and MAGE are in target level indicating Hypoglycemia with high rate of change in glucose. R7 is applicable to the patient when TBR-I, TBR-II, GV are above the target levels, MG, TIR are below the target levels, SDR and MAGE are in target levels indicating Hypoglycemia with high glycemic variability. R8 is applicable to the patient when TBR-I, TBR-II,MAGE are above the target levels, MG,TIR are below the target levels, SDR and GV are in target level indicating Hypoglycemia with high glucose fluctuations.

R9 as Hidden hyperglycemia is applicable when overall glucose metrics i.e., MG,GV,SDR,MAGE,TIR are in target levels but both or either (mentioned as AND/OR in decision rules from Table 2) of TAR -I,TAR-II in the decision rule are above the target glu-cose levels indicating events of hyperglycaemia under normal overall glucose levels. R10 as Hidden hyperglycemia with high rate of change in glucose levels is applicable when overall glucose metrics i.e., MG, GV,MAGE,TIR are in target levels but both or either of TAR -I,TAR-II with SDR are above the target glucose levels. R11 as Hidden hyperglycemia with high glycemic variability in glucose levels is applicable when overall glucose metrics i.e., MG,MAGE,SDR,TIR are in target levels both or either of TAR -I,TAR-II with GV are above the target glucose levels. R12 as Hidden hyperglycemia with high glucose fluctua-tions in glucose levels is applicable when overall glucose metrics i.e., MG,GV,SDR,TIR are in target levels but both or either of TAR -I,TAR-II with MAGE are above the target glucose levels.

R13 as Hidden hypoglycemia is applicable when overall glucose metrics i.e., MG,GV,SDR,MAGE,TIR are in target levels but both or either of TBR -I,TBR-II are above the target glucose levels indicating events of hypoglycaemia under normal overall glucose levels. R14 as Hidden hypoglycemia with high rate of change in glucose levels is applica-ble when overall glucose metrics i.e., MG, GV,MAGE,TIR are in target levels but both or either of TBR -I,TBR-II with SDR are above the target glucose levels. R15 as Hidden hypo-glycemia with high glycemic variability in glucose levels is applicable when overall glu-cose metrics i.e., MG,MAGE,SDR,TIR are in target levels but both or either of TBR -I,TBR-II with GV are above the target glucose levels. R16 as Hidden hypoglycemia with high glu-cose fluctuations in glucose levels is applicable when overall glucose metrics i.e., MG,GV,SDR,TIR are in target levels but both or either of TBR -I,TBR-II with MAGE are above the target glucose levels.

The patient’s condition is assessed with R17 when both or either of TAR -I, TAR-II and both or either of TBR -I TBR-II are above target ranges indicating hyperglycaemia and hypoglycaemia occurring in different intervals in the events of hyperglycaemic spikes and hypoglycaemic episodes [35-37].R18 is assessed to the patient when both or either of TAR -I, TAR-II and both or either of TBR -I TBR-II with SDR are above target ranges indicating occurrence of hyperglycaemia and hypoglycaemia at different intervals with a high rate of change in glucose levels.R19 is assessed to the patient when both or either of TAR -I, TAR-II and both or either of TBR -I TBR-II with GV are above target ranges indicating occurrence of hyperglycaemia and hypoglycaemia at different intervals with a high glycemic variability. R20 is assessed to the patient when both or either of TAR -I, TAR-II and both or either of TBR -I TBR-II with MAGE are above target ranges indicating occurrence of hyperglycaemia and hypoglycaemia at different intervals with a high glucose fluctuations. Each finding differs with a combination of TAR-I, TAR-II, TIR, TBR-I, TBR-II with SDR,GV and MAGE that are above ,below or in the target levels.

However, there is a possibility of a combination of decision rules. When the patient’s condition is a combination of any two conditions between the high rate of change in glucose, high glycemic variability, and high glucose fluctuations, the decision rules are aggregated and displayed, example-Hyperglycemia with a high rate of change in glucose, high glycemic variability is programmed to aggregate as (R2+R3) and displayed.

11.Rule F17 seems to oversimplify complex medical scenarios by covering both hyper and hypoglycemia.

Authors Response: The authors apologize for wrongly writing the statement “Then the glycemic control is unstable in the case of Hyper and Hypoglycemia’ provide an explanation and references supporting for F17.The authors have included the explanation and references in the manuscript.

Authors Action: The authors have updated the manuscript by providing explanation and references in page number 7 (line 322-325) under 2.3 Proposed Decision Rules as,

The patient’s condition is assessed with R17 when both or either of TAR -I, TAR-II, and both or either of TBR -I TBR-II are above target ranges indicating hyperglycemia and hypoglycemia occurring in different intervals in the events of hyperglycaemic spikes and hypoglycaemic episodes [35-37].

12.It is confusing when you write AND/OR without explanation: OR subsumes AND, for one.

Authors Response: The authors apologize for the confusion created due to writing AND/OR and without explanation. The authors have updated the table and provided an explanation in the manuscript.

Authors Action: The authors have updated the manuscript by,

(a) Updating the table in page number 8 (line 341) under 2.3 Proposed Decision Rules as,

Table 2 Decision rules from combined assessment of metrics programmed in ‘CGM Trace’

R.No

Decision Rule for Different Findings

R1

If MG>180 AND TAR -I >25% AND TAR-II >5% AND TIR<70% AND GV>36 AND SDR>5 AND MAGE >40

Then

This is a case of Hyperglycemia with a high rate of change in glucose levels, high glycemic variability, and high glucose fluctuations

R2

If MG>180 AND (TAR -I >25% AND TAR-II >5% AND TIR<70%) AND GV≤36 AND MAGE <40 AND SDR>5

Then 

This is a case of Hyperglycemia with a high rate of change in glucose

R3

If MG>180 AND (TAR -I >25% AND TAR-II >5% AND TIR<70%) AND SDR<5 AND MAGE <40 AND GV>36

Then

This is a case of Hyperglycemia with a high glycemic variability

R4

If MG>180 AND (TAR -I >25% AND TAR-II >5% AND TIR<70%) AND SDR<5 AND GV≤36 AND MAGE >40

Then

This is a case of Hyperglycemia with high glucose fluctuations

R5

If MG<180 AND TBR-I>4% AND TBR-II >1% AND TIR<70% AND GV>36 AND SDR>5 AND MAGE >40

Then

This is a case of Hypoglycemia with a high rate of change in glucose levels, high glycemic variability, and high glucose fluctuations

R6

If MG<180 AND TBR-I>4% AND TBR-II >1% AND TIR<70% AND GV≤36 AND MAGE <40 AND SDR>5

Then 

This is a case of Hypoglycemia with a high rate of change in glucose

R7

If MG<180 AND TBR-I>4% AND TBR-II >1% AND TIR<70% AND SDR<5 AND MAGE <40 AND GV>36

Then 

This is a case of Hypoglycemia with high glycemic variability

R8

If MG<180 AND TBR-I>4% AND TBR-II >1% AND TIR<70% AND SDR<5 AND GV≤36 ANDMAGE >40

Then

This is a case of Hypoglycemia with high glucose fluctuations

R9

If MG<180 AND (TAR -I >25% OR TAR-II >5%)AND TIR>70% AND GV≤36 AND SDR<5 AND MAGE <40

Then

This is a case of Hidden Hyperglycemia

R10

If MG<180 AND (TAR -I >25% OR TAR-II >5%) AND TIR>70% AND GV≤36 AND MAGE <40 AND SDR>5

Then 

This is a case of Hidden Hyperglycemia with a high rate of change in glucose levels

R11

If MG>180 AND (TAR -I >25% OR TAR-II >5%) AND TIR>70% AND SDR<5 AND MAGE <40 AND GV>36

Then

This is a case of Hidden Hyperglycemia with high glycemic variability

R12

If MG>180 AND (TAR -I >25% OR TAR-II >5%) AND TIR>70% AND SDR<5 AND GV≤36 AND MAGE >40

Then

This is a case of Hidden Hyperglycemia with high glucose fluctuations

R13

If MG<180 AND (TBR-I>4% OR TBR-II >1%) AND TIR>70% AND GV≤36 AND SDR<5 AND MAGE <40

Then

This is a case of Hidden Hypoglycemia

R14

If MG<180 AND (TBR-I>4% OR TBR-II >1%) AND TIR>70% AND GV≤36 AND MAGE <40 AND SDR>5

Then 

This is a case of Hidden Hypoglycemia with a high rate of change in glucose levels

R15

If MG<180 AND (TBR-I>4% OR TBR-II >1%) AND TIR>70% AND SDR<5 AND MAGE <40 AND GV>36

Then 

This is a case of Hidden Hypoglycemia with high glycemic variability 

R16

If MG<180 AND (TBR-I>4% OR TBR-II >1%) AND TIR>70% AND SDR<5 AND GV≤36 AND MAGE >40

Then

This is a case of Hidden Hypoglycemia with high glucose fluctuations

R17

If (TAR -I >25% OR TAR-II >5%) AND (TBR-I>4% OR TBR-II >1%) AND TIR<70%

Then

This is a case of Hyperglycemia and Hypoglycemia at different intervals.

R18

If (TAR -I >25% AND TAR-II >5%) AND (TBR-I>4% ORTBR-II >1%)AND TIR<70 AND GV≤36 AND MAGE <40 AND SDR>5

Then 

This is a case of Hyperglycemia and Hypoglycemia at different intervals with a high rate of change in glucose levels

R19

If (TAR -I >25% OR TAR-II >5%) AND (TBR-I>4% AND TBR-II >1%)AND TIR<70 AND SDR<5 AND MAGE <40 AND GV>36

Then 

This is a case of Hyperglycemia and Hypoglycemia at different intervals with high glycemic variability 

R20

If (TAR -I >25% OR TAR-II >5%)AND (TBR-I>4% AND TBR-II >1%)AND TIR<70 AND SDR<5 AND GV≤36 AND MAGE >40

Then

This is a case of Hyperglycemia and Hypoglycemia at different intervals with high glucose fluctuations

Abbreviations: R.No, Rule number; MG, mean glucose; GV, glucose variability; TIR, time in range; TAR, time above range; TBR, time below range; SDR, standard deviation rate of change; MAGE, mean amplitude of glycemic excursions                      

(b) Providing an explanation on AND/OR in page number 7 (line 297 to 300) under 2.3 Proposed Decision Rules as,

R9 as Hidden hyperglycaemia is applicable when overall glucose metrics i.e., MG, GV, SDR, MAGE, TIR are in target levels but both or either (mentioned as OR in decision rules from Table 1) of TAR -I, TAR-II in the decision rule are above the target glucose levels indicating events of hyperglycaemia under normal overall glucose levels.

13.In rule F17, scientific supports required for this assumption (hyper and Hypoglycemia at the same time). Authors also have similar claim in line 432. The publication referred below ssupoports coexistance of "rebound" hypo- and hyper-glycemia. However, it doesnt support "concurrent" occurrence of those phenomenon,. In other words, a hypoglycemia can be followed by hyperglycemia (or vice versa), but they are not happening simultaneously.

Hansen KW, Bibby BM. Rebound Hypoglycemia and Hyperglycemia in Type 1 Diabetes. J Diabetes Sci Technol. 2023 May 3:19322968231168379. doi: 10.1177/19322968231168379. This assumption may lead to false results.

Authors Response: The authors apologize for missing out to mention the reference for F17 to F20.The authors have included the references in the manuscript.

Authors Action: The authors have updated the manuscript by adding the references in page number 7 (line 322 to-line 325) under 2.3 Proposed Decision Rules as,

The patient’s condition is assessed with F17 when both or either of TAR -I, TAR-II and both or either of TBR -I TBR-II are above target ranges indicating hyperglycaemia and hypoglycaemia occurring in different intervals in the events of hyperglycaemic spikes and hypoglycaemic episodes [35-37].

14.Line 263: It is worth to mention link of code repo/webpage, if this project/page is publicly available.

Authors Response: The authors are thankful to the esteemed reviewers for their remark on code repo/webpage link.

                                The authors want to state that the webpage is delivered on local host server. Global server requires funds for deployment and maintenance which is considered as a future work. The code repo/webpage of the project will be accessible after deploying in global server once we get the funds.

15.Table 4: manova typically provides a single p-value related to the overall test, not individual p-values for each combination of condition and metric.

Authors Response: The authors apologize for wrongly writing the p-value from Table 3. The authors have corrected the legend of Table 3 and added the p-value which is considered from Box’s M test.

Authors Action: The authors have updated the manuscript by

(a) changing the legend of Table 3 in page number 10 (line 355) under 3.1. Validation of Decision Rules as,

Table 3 Assumptions and p_value from MANOVA one-way test

No.

Assumptions

Test performed

Transformation  performed

p_value

1

Normality Distribution

Normality test

Inverse DF

    -

2

Homogeneity of covariance matrices

Box’s M test

      -

0.531

3

Effect of independent variables on dependent variables

Tests of Between-Subjects Effects

      -

0.791

4

Overall significance

Wilk’s Lambda

      -

0.001

Abbreviations: No, number; DF, distribution function

(b) Adding the p_value from overall test in page number 11 (line 377 to 380) under 3.1. Validation of Decision Rules as,

The overall significance obtained between the Findings and the metrics was p=0.001 from Wilk’s lambda test where partial eta squared=0.395 was acquired. It can be interpreted that there is a difference and a strong significance between the findings and the metrics as p<0.05 rejecting the null hypothesis.

16.Line 296: Which software used to run this tests? readers could also benefit from normality check of data, or if any correction conducted whit MANOVA.

Authors Response: The authors apologize for missing out to mention about the software used to run the MANOVA test, normality check of data and correction conducted. The authors have added the details regarding software used to run the MANOVA test, normality check of data and correction conducted in the manuscript.

Authors Action: The authors have updated the manuscript by

(a) Mentioning the software used for running MANOVA test in page number 10 (line 359) under 3.1. Validation of Decision Rules as,

Statistical analysis using one-way MANOVA was performed using IBM SPSS Statistics 27 software.

(b) Explaining the normality check of data and transformation applied in page number 10 (line 360 to 380) under 3.1. Validation of Decision Rules as,

The normality of data is the fundamental assumption of MANOVA. The assumptions and p_value from MANOVA one-way test are described in Table 3. As the data was skewed, an inverse distribution function (DF) was performed to transform data into a normal distribution. The second assumption of MANOVA is that the covariances of matrices must be equal which was achieved with p=0.531(rejecting null hypothesis that covariances of matrices are not equal). The findings were the factor with 20 levels i.e., F1-F20 which are the independent variables. The dependent variables were the metrics i.e., SDR, MG, GV (%), TIR (%), TAR-I (%), TAR-II (%), TBR-I (%), TBR-II (%), and MAGE. From tests of between-subjects effects, p=0.791 was achieved which stated that the independent variables affected dependent variables (rejecting the null hypothesis that there is no effect of independent variables on dependent variables) satisfied the third assumption of MANOVA.CONGA and MODD are omitted from the decision rules as they don’t have thresholds. However, SDR is chosen over SD due to its asymmetric nature with unequal distribution of hypoglycemic and hyperglycemic ranges leading towards biased assessment. SD, CONGA, and MODD are considered as supporting metrics for data analysis. A null hypothesis was formed stating that the Findings would not differ and have no significance among all the dependent metrics. The overall significance obtained between the Findings and the metrics was p=0.001 from Wilk’s lambda test where partial eta squared=0.395 was acquired. It can be interpreted that there is a difference and a strong significance between the findings and the metrics as p<0.05 rejecting the null hypothesis.

17.It would be helpful to describe how each label (F1-20) assigned to each patient, before conducting MANOVA. Was this based on the opinion of a clinician?

Authors Response: The authors apologize for the confusion caused due to wrong organization of ‘Validation of Decision Rules’ after Table 3 CGM data analysis from ‘CGM Trace’- Public dataset from JCHR-JAEB.

                                          The authors have first validated the decision rules using MANOVA and based on the validation, (F1-20) were assigned to each patients. This was based on opinion of the clinician. (F1-20) are also updated as (R1-20) to avoid confusion.

Authors Action: The authors have updated the manuscript by

(a) Replacing the F1-F20 as R1-R20 (rule 1-rule 20) to avoid confusion.

(b) Re-organizing the 3. Results section by making the ‘Validation of Decision rules’ as 3.1. Validation of Decision Rules in page number 10 (line 353) and 3.2. Web Interface and Data Analyzation from ‘CGM Trace'.

18.Line 314: You can simply name tables in supplementary material as S!, S2, and avoid repeating "supplementary material" and confusing reader with Tables inside main text.

Authors Response: The authors apologize for causing confusion due to the repetition of " Table 1 from Supplementary Material" throughout the manuscript. The authors have renamed the table in supplementary material and highlighted in the manuscript.

Authors Action: The authors have updated the manuscript by,

(a) naming the Table in supplementary material as S1 and mentioning “Table 1 from Supplementary Material “as S1 for the first time in the manuscript in page number 10 (line 351) under 3. Results as,

37 patients from public data are provided in S1, Supplementary Material.

(b) updating the “Table 1 from Supplementary Material” as “S1” throughout the manuscript in yellow highlights.

19.Line 325 and later: "Finding N from Table 1 .... is applicable as a decision rule" repetition could decrease readability of the section. This sentence has been repeatedly used for each finding.

Authors Response: The authors apologize for the inconvenience caused due to repeatability of the sentence "Finding N from Table 1 .... is applicable as a decision rule". The authors have reduced the repetition by classifying the decision rules into hyperglycemia, hypoglycemia, hidden hyperglycemia, hidden hypoglycemia, hyperglycemia and hypoglycemia, and in writing the findings.

Authors Action: The authors have reduced the usability of the sentence and highlighted throughout the section in page number 13 (lines 426 to 563) under 3. Results

20.Line 328-329: Is the combination of decision rules into a single finding is based on another "higher level" rule? When system is allowed to aggregate findings? A description of this process in methods and material section would be helpful.

Authors Response: The authors apologize for not mentioning about the aggregation of findings. The authors have added the description of the process in methods and material section.

Authors Action: The authors have updated the manuscript in page number 7 (line 336 to 340) under 2.3 Proposed Decision Rules by,

However, there is a possibility of combination of decision rules. When the patient’s condition is a combination of any two conditions between high rate of change in glucose, high glycemic variability and high glucose fluctuations, the decision rules are aggregated and displayed, example-Hyperglycemia with a high rate of change in glucose, high glycemic variability is programmed to aggregate as (R2+R3) and displayed.

21.Line 432: those statements could help earlier, plus it helps to use a reference, e.g.: Hansen KW, Bibby BM. Rebound Hypoglycemia and Hyperglycemia in Type 1 Diabetes. J Diabetes Sci Technol. 2023 May 3:19322968231168379. doi: 10.1177/19322968231168379.

Authors Response: The authors are thankful to the esteemed reviewers for their remark on adding a reference. The authors have added the reference as suggested.

Authors Action: The authors have added the reference as [37] and updated the manuscript in page number 21 (line 795) as,

  1. Hansen KW, Bibby BM. Rebound Hypoglycemia and Hyperglycemia in Type 1 Diabetes. J Diabetes Sci Technol. 2023 May 3:19322968231168379. doi: 10.1177/19322968231168379. Epub ahead of print. PMID: 37138541.

22.Line 549: GP is generally made to present overview of changes. Then clinician could go back to daily "mean glucose" chart to check variability of glucose levels. I recommend to elaborate more over the difference of your tool versus other similar tools.

Authors Response: The authors are thankful to the esteemed reviewer for their remark on elaborating on the difference of the proposed tool versus other similar tools.

                                  The authors mentioned about the advantage of ‘CGM Trace’ in lines 563 to 569 previously but did not include the statement about how the proposed tool differs from other tools. The authors have now added few lines before the paragraph and updated the manuscript.

Authors Action: The authors have updated the manuscript by elaborating over the difference ‘CGM Trace’ as a tool is when compared to other tool in page number 19 (line 643 to 667) under 4. Discussion as,

The ’CGM Trace’ differs from AGP report by combining statistical metrics at a single place and providing ‘Findings’ for 20 cases. Each ‘Finding’ defines a clear status of patient’s glucose levels which can be accessed anytime. The glucose profile from ‘CGM Trace’ can be saved during every visit and assessed to create a tailored treatment plan which is lacking in AGP report. ‘CGM Trace’ enhances the AGP report by providing insights into patients’ complete glucose profile, interday GV, intraday GV, hidden, and temporal patterns of glucose fluctuations from combined metrics and graphical trends for any selected period which lacks in AGP report. The doctor can examine the metrics, trends, and decision rules to create treatment adjustments and strategies in case of a rise in glucose levels, post-meal spikes, midnight lows, glucose fluctuations, GV patterns, response to medications, causes, duration, frequency, severity, and recurrence of hyperglycemia and hypoglycemia. Interpretation of day-to-day or half-day inconsistencies is straightforward in ‘CGM Trace.’ Minute data details below 10% of the lowest glucose levels can be visualized from intraday, hyperglycemic, and hypoglycemic graphs lacking in the AGP report. A report can be generated from the ‘CGM Trace’ for maintaining records. ‘Glucose Profiles’ are a rapid method to analyze the glucose trends/graphs ranging from 1 day to n number of days proving to be a valuable tool during the emergency condition of the patient. Response to medication, persistence of glycemic peaks or lows, reoccurrences of hyperglycemia and hypoglycemia can be analyzed visually from ‘Glucose profiles’ of ‘CGM Trace’ which lacks in AGP report.This work showcases the potential of a combinational assessment of metrics and respective findings as a decision-support tool to enhance the AGP report. The findings highlight the potential of identifying many combinational cases of hyperglycemia and hypoglycemia which indicate underlying diseases. The management of glycemic variability, fluctuations, and hidden patterns improves overall glucose control for a better lifestyle and diabetic management by ‘CGM Trace’.

23.Line 553: Why access to: “glucose profiles" should be limited to only emergency cases?

Authors Response: The authors apologize for the confusion created with the statement “In the cases of emergency, the doctor can access the ‘Glucose Profiles’ for varied summaries on glucose trends that can be accessed from a min of 1 day to n number of days.”

                                  The authors did not mean that the “glucose profiles” are limited to emergency cases. ‘Glucose profiles’ are the different graphs which can be accessed quickly in case of emergency. The authors have rewritten the statement in the manuscript.

Authors Action: The authors have updated the manuscript by rewriting the full sentence in page number 19 (line 657) 4. Discussion as,

‘Glucose Profiles’ are a rapid method to analyze the glucose trends/graphs ranging from 1 day to n number of days proving to be a valuable tool during the emergency condition of the patient. Response to medication, persistence of glycemic peaks or lows, reoccurrences of hyperglycemia and hypoglycemia can be analyzed visually from ‘Glucose profiles’ of ‘CGM Trace’ which lacks in AGP report.

24.Line 56-562: Repetition of exclusion criteria

Authors Response: The authors apologize for repeating the exclusion criteria. The authors have removed the exclusion criteria from the manuscript.

Authors Action: The authors have removed the repetition of exclusion criteria and updated the manuscript in page number 19 (line 668) under 4. Discussion as,

Despite the findings, there are limitations in the proposed work. The proposed work is implemented on 67 patients. It is important to acknowledge that the results therefore may not be directly transferable to the patient datasets of exclusion criteria.

25.Future work pointers are weak, how these findings can help further studies?

Authors Response: The authors are thankful to the esteemed reviewers for their remark on future work. The authors have included the future work from the findings in the manuscript.

Authors Action: The authors have updated the manuscript by adding the future work pointers from findings in page number 19 (line 675 to 691) under 4. Discussion as,

enhanced by considering a large number of datasets with coexisting conditions, finding correlations between the variables and if the existing conditions effect the variables among patients’ demographics will be taken as a future work. Another limitation of the proposed work is that the user must have an internet connection to access the webpage. As a future work, the webpage will be developed into a mobile application (app) for ease of access and deployed on a global server. The User-friendly interface of the app will be crafted to avail maximum benefit from the app to rural and urban people. The design will consider literacy levels, language preferences, and device compatibility. Accessibility of important features in offline mode and training programs by collaborating with local healthcare workers in rural areas for education about the app will be considered for the feasibility of the technology. The findings during every doctor visit from each patient can be gathered and integrated with machine learning algorithms for prediction and treatment recommendations or better diabetic management. The findings can help future researchers to develop optimal glycemic control strategies. The efficacy of new drugs can be assessed by identifying their response from findings can be considered as future work.

Round 2

Reviewer 1 Report (New Reviewer)

Comments and Suggestions for Authors

The authors have sufficiently addressed all the comments and hence, I recommend that this manuscript be accepted for publication in the current state.

Reviewer 2 Report (New Reviewer)

Comments and Suggestions for Authors

All my comments were attempted to be answered

Comments on the Quality of English Language

Passable.

This manuscript is a resubmission of an earlier submission. The following is a list of the peer review reports and author responses from that submission.

Round 1

Reviewer 1 Report

Comments and Suggestions for Authors

Thanks for giving me the opportunity to read your proposal in the enhancement of AGP profile to assist the physicians in their decision for adapting the treatment to type 1 diabetic patients. 

General comments: It is difficult to understand how the web interface “CGM trace” will be more helpful for diabetologists or nurses or patients for improving the management of diabetes. The proposal lies on 10 “new” metrics already published by other authors, and the input of the authors herein is to incorporate statistical metrics, SDR (that seems already proposed by David Rodbard in ref 13 Czerwoniuk D et al 2011), and IQR of glucose values (already cited in ref 13). So, what is really the novelty in this paper? In addition, my main concern is about the existence of a consensus for the validation of all these proposed “new” metrics. Indeed, to my knowledge, the only glucose metrics validated with clear recommendations by the ATTD (Battelino T et al. 2019) and used in routine by diabetologists are TIR 70-180 mg/dL that should be >70% (or > 50% for older patients), TBR 69-54 mg/dL (<4% or 1% if older), TBR<54 mg/dL (<1%), TAR 181-250 mg/dL (<25%), TAR>250 mg/dL (<5%), CV% (≤36%).

Regarding SDR you are proposing, I did not find in ref 13 the indication of the threshold value you have mentioned in your paper, i.e., ≥5. In clear, who did validate all these new metrics through a large consensus study, and set-up their threshold values? Only clear consensus can be used for giving medical decision. So, the objective of your paper is not reachable through the analysis done with 10 patients (listed in Table 2). 

Specific comments:

1- Introduction: there are Imprecisions and abbreviations are not always indicated that makes the text sometimes difficult to read. As for example, BG (Blood Glucose I guess). Please note that AGP profile gives “interstitial glucose values” and not “blood value”. TBR is “time below range” of interstitial glucose values and not “time below hypoglycemia” and is indicator of hypoglycemia. As well, TAR is “time above range” and is indicator of hyperglycemia. All targets are for “interstitial glucose values” and it is assumed there is some correspondence with blood glucose. So, it is an estimation of glycemia, not real glycemia. 

2- Materials and Methods: 

- The clinical centers you have indicated for the study whose you have used the available glucose metrics data (from JCHR-JAEB center) are not well indicated (names are not all exact). I went on their website, and it is indicated that the persons who will use their data have to cite the name (i.e., entire title) of the study, and this is missing herein. Please complete the information as requested by the JCHR-JAEB center.  

- Figure 1: why did you use 50% as threshold value for TIR, 33% for GV, 0 for TBR and TAR? This is not clear and different than the recommendations from the ATTD. What is a low or high MOOD, MAGE, CONGA (i.e., what are the threshold values)? 

- 2.2 New metrics: the term “treatment titrations” is unclear, please explain or modify (did you mean insulin doses?). Hypo and hyperglycemic markers are already indicated in AGP profile; for an indicator of hypoglycemia, TBR 69-54 mg/dL and TBR <54 mg/dL are used, and for hyperglycemia TAR 181-250 mg/dL and TAR>250 mg/dL (sometimes TAR>320 mg/dL) are used. What are exactly your proposals in points 3 and 4? This is not clear. 

- Table 1 is difficult to read. 

- Statistics: the way the analysis of the data was done is not clearly indicated. ANOVA for comparing what with what? Did you build several models? 

3- Results: 

- The data set you have used is from at least 50 patients with type 1 diabetes. Why did you use only the data of 10 patients in Table 2 (with the other data in supplementary Table)? How did you select the 10 patients of the Table2? 

- Figure 2: the legend of the figure must indicate whether it is glucose distribution on a specific 24 hours or a mean of 14 days? Please clarify. 

- Figure 3: some data are missing to have a complete 24 hours. Fig 3 part C is about glucose values at night, but the night starts at 5 PM and finished at 9PM. This is unclear. 

- Many results are in the discussion section. Please move them to the Results section. 

- Explain what a “hidden hyperglycemia” is. 

- Finding 1: Patient 6 does not have a TIR<50% (it is 58.7), his/her SDR is <5 (not>5 as indicated in the text) and MG is <180 mg/dL (not> as indicated in the text). Please correct. Please verify all your data indicated in the text, other errors might be found.

4- Discussion: to be rewritten. 

5- Conclusion: I am not convinced all that can help further the diabetologist to take decision for precision medicine. The proof has to be done through a clinical study. 

Comments on the Quality of English Language

None

Reviewer 2 Report

Comments and Suggestions for Authors

The authors present new ideas for analyzing CGM profiles, incorporating a combination of old and new CGM metrics.

Although the approach is interesting, the whole is very confused and resembles a personal opinion with no real scientific rationale clearly demonstrated. 

There are numerous inaccuracies throughout the manuscript, such as:

- the CGM does not give blood glucose values, but interstitial glucose values

- several copy/paste errors in the "new metrics" section

- standard deviation should not be expressed as a percentage (confusion with CV)

- MODD is not an assessment of intra-day but inter-day variability.

- and so on...

There are many bibliographical references that are either inappropriate or not cited in the right place. For example, in the introduction, T Battelino's international AGP consensus should be cited. Regarding CV, the work of L Monnier should be cited.

Finally, the conceptual data developed on the new metrics are sometimes mixed with comments concerning the impact of this or that metric on the risk of complication, for example... these different mixed concepts contribute to making the whole really confusing.

Comments on the Quality of English Language

Quality of english is not an issue
